# Lipid-droplet associated mitochondria promote fatty-acid oxidation through a distinct bioenergetic pattern in male Wistar rats

Noble Kumar Talari[1], Ushodaya Mattam[1], Niroj Kumar Meher [1],
Arun Kumar Paripati[1], Kalyankar Mahadev[2], Thanuja Krishnamoorthy[3] &
Naresh Babu V. Sepuri [1] ✉

Mitochondria empower the liver to regulate lipid homeostasis by enabling fatty acid oxidation during starvation and lipogenesis during nutrient-rich conditions. It is unknown if mitochondria can seamlessly regulate these two distinct processes or if two discrete populations of mitochondria achieve these two functions in the liver. For the first time in the liver, we report the isolation of two distinct populations of mitochondria from male Wistar rats on an *ad-libitum* diet: cytoplasmic mitochondria and lipid droplet-associated mitochondria. Our studies show that while lipid droplet mitochondria exhibit higher fatty acid oxidation and are marked by enhanced levels of pACC2, MFN2, and CPT1 activity, cytoplasmic mitochondria are associated with higher respiration capacity. Notably, lipid droplet-associated mitochondria isolated from a non-alcoholic fatty liver disease (NAFLD) rat model are compromised for fatty acid oxidation. We demonstrate the importance of functional segregation of mitochondria as any aberration in lipid droplet-associated mitochondria may lead to NAFLD.

The liver plays a central role in fatty acid metabolism. Fatty acids accumulate in the liver by traveling through the hepatocellular uptake system; notwithstanding, *de novo* biosynthesis of fatty acids also occurs in the liver. The liver combats this by oxidizing the fatty acids or secreting them as triglycerides associated with very low-density lipoproteins. However, when the liver is infused with excess intake of fatty acids, the liver tries to maintain the fatty acid level by conserving them as fat in the adipose tissue, leading to obesity, mainly exemplified as fat around the waist. Obesity-linked insulin resistance predicates the onset of metabolic syndrome, which includes a cluster of conditions like type 2 diabetes mellitus, atherosclerosis, hypertension, and a propensity to hepatic steatosis (fatty liver) progression[1]. Excessive fat in the liver can ultimately lead to fatty liver disease (FLD). Though FLD is clinically categorized into alcoholic FLD (AFLD) and non-alcoholic FLD (NAFLD), the histological hallmarks are indistinguishable. They include hepatic steatosis and steatohepatitis, progressing to liver cirrhosis and hepatocellular carcinoma. Obesity and FLD are rising, including NAFLD, and are now considered growing global health problems[2]. Unraveling the etiology and pathology of NAFLD is required to improve the prognosis and develop novel therapeutics.

Inter-organelle interactions have been correlated with diverse cellular functions, including adaptive cellular response to the nutrient environment[3–5]. Each organelle has a characteristic distribution and dispersion, which is dynamic. Liver lipid droplets (LDs) are highly

[1]Department of Biochemistry, School of Life Sciences, University of Hyderabad, Hyderabad TS-500046, India. [2]School of Medical Sciences, University of Hyderabad, Hyderabad TS-500046, India. [3]Vectrogen Biologicals Private Limited, BioNEST, School of Life Sciences, University of Hyderabad, Hyderabad TS-500046, India. ✉e-mail: nareshuohyd@gmail.com

dynamic organelles enriched in neutral lipids at the core, while their outer phospholipid layer is studded with integral and peripheral proteins. Gathering evidence highlights the importance of mitochondria-LD association for regulating energy homeostasis and maintaining a balance between the biogenesis of LD and lipolysis. Conditions like nutrient stress, starvation, and thermogenesis trigger the translocation of fatty acids from LD into mitochondria for the β-oxidation to proceed[6,7]. Paradoxically, mitochondria-LD contact is also responsible for enhancing LD biogenesis to shield the mitochondria from lipid-induced toxicity[8–11].

Under healthy conditions, the liver regulates various aspects of lipid metabolism, taking cognizance of the nutritional status of the cell. A balance must exist between fatty acid uptake, synthesis of fatty acid, partitioning intracellular lipids into storage, and oxidation of fatty acids. Mitochondria can support fatty acid synthesis or oxidation by regulating fatty acid entry via the carnitine palmitoyltransferase (CPT1/2) system[12,13]. Increased malonyl CoA production blocks CPT1, an inhibition that favors fatty acid synthesis. This understanding revealed that mitochondria could support lipid synthesis or oxidation at a time

but not both. However, recent studies have challenged this concept. Both brown adipose tissue and immune cells can accommodate these processes simultaneously[8,14–16].

A recent study has shown that mitochondria associated with LDs in brown adipose tissue exhibit increased pyruvate oxidation, electron transport, and ATP synthesis besides supporting LD biogenesis. Curiously, the mitochondria depart from the LDs when β-oxidation is activated in brown adipose tissue[8]. In contrast to studies in brown adipose tissue, not much is known about mitochondria associated with LDs in the liver. The main reason is the lack of a suitable method to isolate mitochondria associated with LDs from the liver. This study tries to delineate LD-associated mitochondria (LDM) functions in the rat liver using cytoplasmic mitochondria (CM) as the benchmark. We developed a method to isolate LDMs from rat liver to enable this study. Next, using a combination of techniques, including Transmission Electron Microscopy (TEM), fluorescence, and biochemical assays, we studied LDMs from the liver. Our studies reveal that LDMs in the liver are segregated from the CM to perform specialized functions even under an *ad libitum* food condition. The liver LDMs are associated with

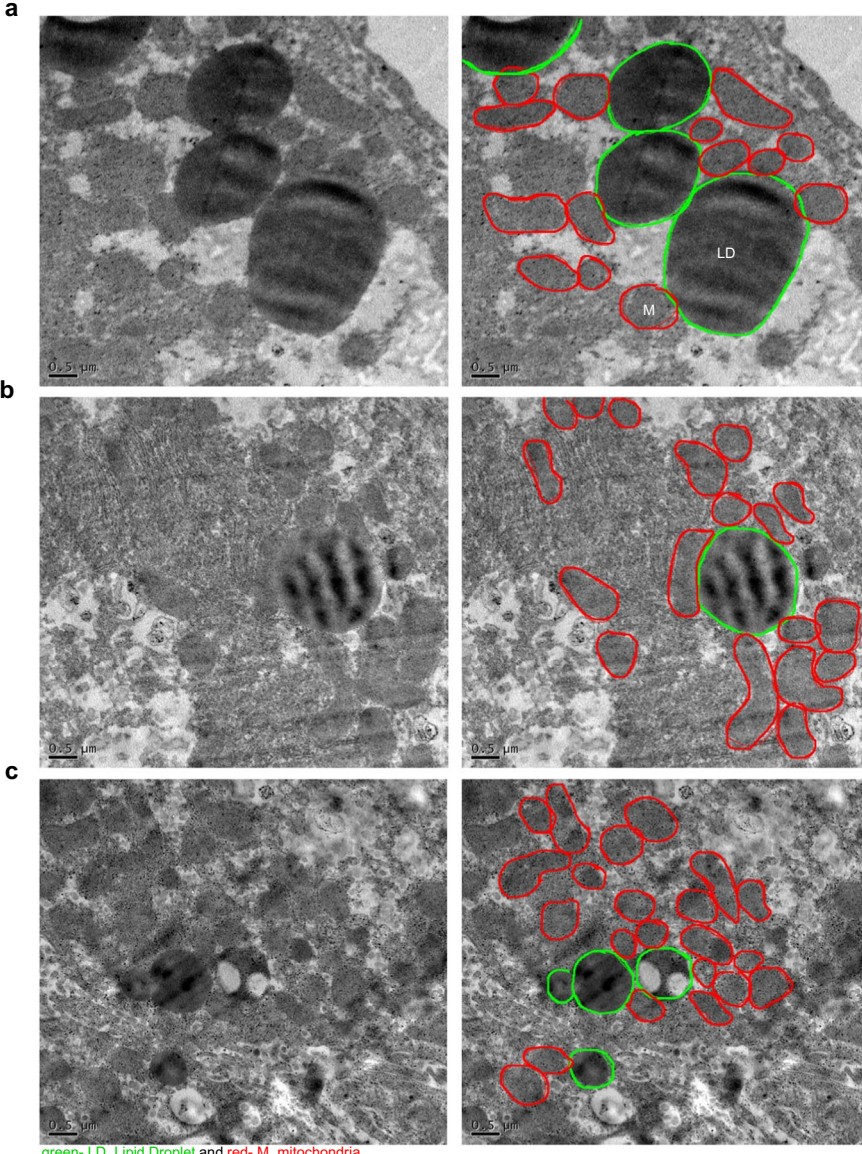

green- LD, Lipid Droplet and red- M, mitochondria

**Fig. 1 | Transmission Electron Microscopy (TEM) images of Lipid Droplet associated Mitochondria (LDM) from rat liver.** Representative TEM images of rat liver harvested from adult Wistar rats (**a**–**c**; *n* = 3) maintained at room temperature with *ad-libitum* food show Lipid Droplets (LD) in contact with mitochondria (scale bar, 0.5 μm). All the right panels have a green line tracing the LDs and a red line outlining the mitochondria.

increased fatty acid oxidation (FAO) and reduced respiration with specific bioenergetic outcomes. Most notably, in the high fat diet induced NAFLD rat model, LDM exhibits decreased FAO underscoring the physiological significance of the liver LDM.

## Results

### LDs are associated with mitochondria in the liver

The LD-mitochondria association is critical in lipid homeostasis among the various inter-organelle interactions. The association of mitochondria with LD has been observed in different tissues[7,9,17–19]. However, there is a lacuna in their association in the liver, the primary organ in fatty acid metabolism. This dearth of information has been attributed to the lack of a suitable method to isolate LDMs from the liver. Before developing a method for isolating LD-mitochondria from rat liver, we first studied the association of LD-mitochondria by TEM.

We isolated liver from adult Wistar rats ($n$= 3) maintained at room temperature under an *ad libitum* (normal and unrestricted) food condition. As discussed in the methods section, we performed whole animal perfusion with 4% PFA and 1% glutaraldehyde to fix the liver tissue for TEM studies. Our electron micrographs show that LDs are in close contact with mitochondria (Fig. 1). A closer analysis of the TEM images reveals that LDs are surrounded by a larger cluster of mitochondria. Curiously, liver can respond to nutrient starvation either by triggering LD hydrolysis or gluconeogenesis[20,21]. During LD hydrolysis, larger LDs break away into smaller round LDs that are preceded by the translocation of FA from LD to mitochondria[6,7]. In contrast, another study in brown adipose tissue has suggested that instead of LD hydrolysis, mitochondria bound to LDs support their expansion by providing ATP for triglyceride synthesis[8]. In the case of the liver, it is not known if the association of LDs with mitochondria is a harbinger of LD hydrolysis or LD expansion.

### Isolation of LD associated mitochondria from liver by differential centrifugation

To test if the association of LDs with mitochondria leads to LD hydrolysis or expansion, we first isolated the LDMs from rat livers. Low-speed centrifugation was recently used to separate the fat layer in brown adipose tissue, followed by high-speed centrifugation for isolating LDMs. However, it is arduous to follow this method for separating liver fat. We have developed a modified method to isolate liver LDM from rat liver, as described in the methods section. After low-speed centrifugation (900 × g), we separated the supernatant fraction and layered it with buffer B. Next, centrifugation (2000 × g) is carried out to isolate the fat layer. This method was previously used to isolate crude LD from mouse liver[22]. The fat layer is subjected to high-speed centrifugation (10400 × g) to isolate LDM (Fig. 2a and S1). The supernatant fraction beneath the fat layer from 2000 × g centrifugation is subjected to high-speed centrifugation (10400 × g) to isolate CM (Figs. 2a and S1).

The fat layer collected after 2000 × g centrifugation was co-stained with the neutral lipid dye BODIPY (493 nm/503 nm) and mitochondrial dye MitoTracker red (Fig. 2b) to check if it is enriched for LDMs. Our confocal microscopic images show that 95% of LD-Mitochondria contacts have been preserved in the fat layer after low-speed centrifugation (Fig. 2b). The chance of CM contamination is minimal as there is a 3 ml barrier in the form of buffer B between the fat layer enriched with LDM and the CM-containing supernatant fraction. As we could not separate LDs from mitochondria by differential centrifugation, we resorted to mild vortexing to disrupt the LD-mitochondrial contacts. After high-speed centrifugation, the stripped fat layer hardly fluoresces with MitoTracker but stains for BODIPY (Fig. 2c, d), suggesting that the fat layer has been effectively stripped of mitochondria but has LDs, albeit smaller in size. While LDs are fragile and disintegrate into smaller LDs, LDMs are resistant to mild vortexing. Interestingly, the LDM pellet displays minor BODIPY staining in juxtaposition but good staining with MitoTracker (Fig. 2e), confirming that the sample is enriched for LDM.

Quantification of the LD content in the fat layer before stripping and in the LDM pellet revealed that the latter had 95% lower LD content than the fat layer (Fig. 2f, g). The results mentioned above suggest that using the modified method, one can cleanly separate and isolate LDM from adult rat liver. In addition, we have also probed the LDM and CM samples with other antibodies for detecting the presence of endoplasm reticulum (GRP78 and calreticulin), lysosomes (lamp1), and cytoplasm (GAPDH) (Fig. S2c). Both CM and LDM show the presence of ER, while mild lysosomal presence is observed in the LDM sample. However, CM and LDM are negative for cytoplasm (Fig. S2c).

### Liver LDMs are specialized for high FAO

A delicate balance between FA synthesis and oxidation is essential for LD homeostasis. We measured FAO capacity to examine if the association of LDs with mitochondria leads to LD hydrolysis or expansion. Immediately after isolation, rat liver samples enriched for LDM and CM were subjected to FAO assay (Abcam ab222944) (Fig. 3a). The assay was performed according to the manufacturer's protocol. We measured the fatty acid-driven oxygen consumption by supplying 18 C Oleic Acid (OA) as a substrate (Fig. 3a) after ensuring that an equal amount of mitochondrial protein was present in both samples (Fig. 3e).

A two-fold increase in FAO is observed in the LDM sample compared to CM when OA is used as a substrate (Fig. 3a). This result was further confirmed when OA + FCCP (maximal) substrate also increased FAO significantly in the LDM sample (Fig. 3a). To explain the increased FAO in LDM compared to CM, we reasoned that this augmentation might be attributed to the increased activity of carnitine palmitoyltransferase 1 (CPT1), a rate-limiting enzyme in the FAO pathway. CPT1 controls the translocation of FA from the cytosol to the mitochondria. We examined the protein levels of CPT1 in LDM and CM by western blot analysis. Unexpectedly, we did not observe any changes in the CPT1 levels (Fig. 3b, c). Therefore, we hypothesized that CPT1 activity is higher in LDM compared to CM. To test if this is true, we monitored CPT1 activity (Fig. 3d). Consistent with our hypothesis, we find that CPT1 activity is greater in LDM than in CM and may partly explain the increased FAO observed in LDM.

Malonyl CoA is another crucial determinant for the rate of FAO as it regulates the entry of FA into mitochondria by regulating CPT1[23]. Malonyl CoA concentrations regulate the switch between fatty acid synthesis and oxidation. Malonyl CoA is produced from acetyl CoA via the enzymatic action of acetyl CoA carboxylase (ACC). In the liver, ACC exists in two isoforms; ACC1, localized to the cytosol, and ACC2, which is present in the mitochondria[24]. Significantly, the sub-cellular localization of ACC determines whether it is associated with the synthesis or oxidation of FA. Malonyl CoA synthesized via the ACC1 path is involved in FA synthesis, while malonyl CoA synthesized by the action of ACC2 regulates the carnitine palmitoyl CoA shuttle system[24]. It has been shown that depletion of ACC2 in mice increases mitochondrial FAO[25]. Hence, we looked at the phosphorylation status of ACC2, which is correlated with its inactivation. Western blot analysis shows that the phosphorylation of ACC2 (p-ACC) is relatively higher in liver LDM samples than in CM samples, with no change in total ACC levels (Fig. 3b, c). Based on the above results, we conclude that liver LDM is specialized for high FAO and this is correlated with high pACC2 and CPT1 activity under an *ad libitum* food condition.

### LDM is bioenergetically differentiated

As reducing equivalents from both FAO and TCA cycles enter the electron transport chain (ETC), and the oxygen consumption readout for FAO using OA is high, we monitored respiration using TCA cycle substrates. Respiration was monitored in LDM and CM samples using glutamate/malate and succinate as substrates. An equivalent amount of LDM and CM samples were subjected to an oxygen consumption assay. The basal state II respiration is relatively higher in CM compared

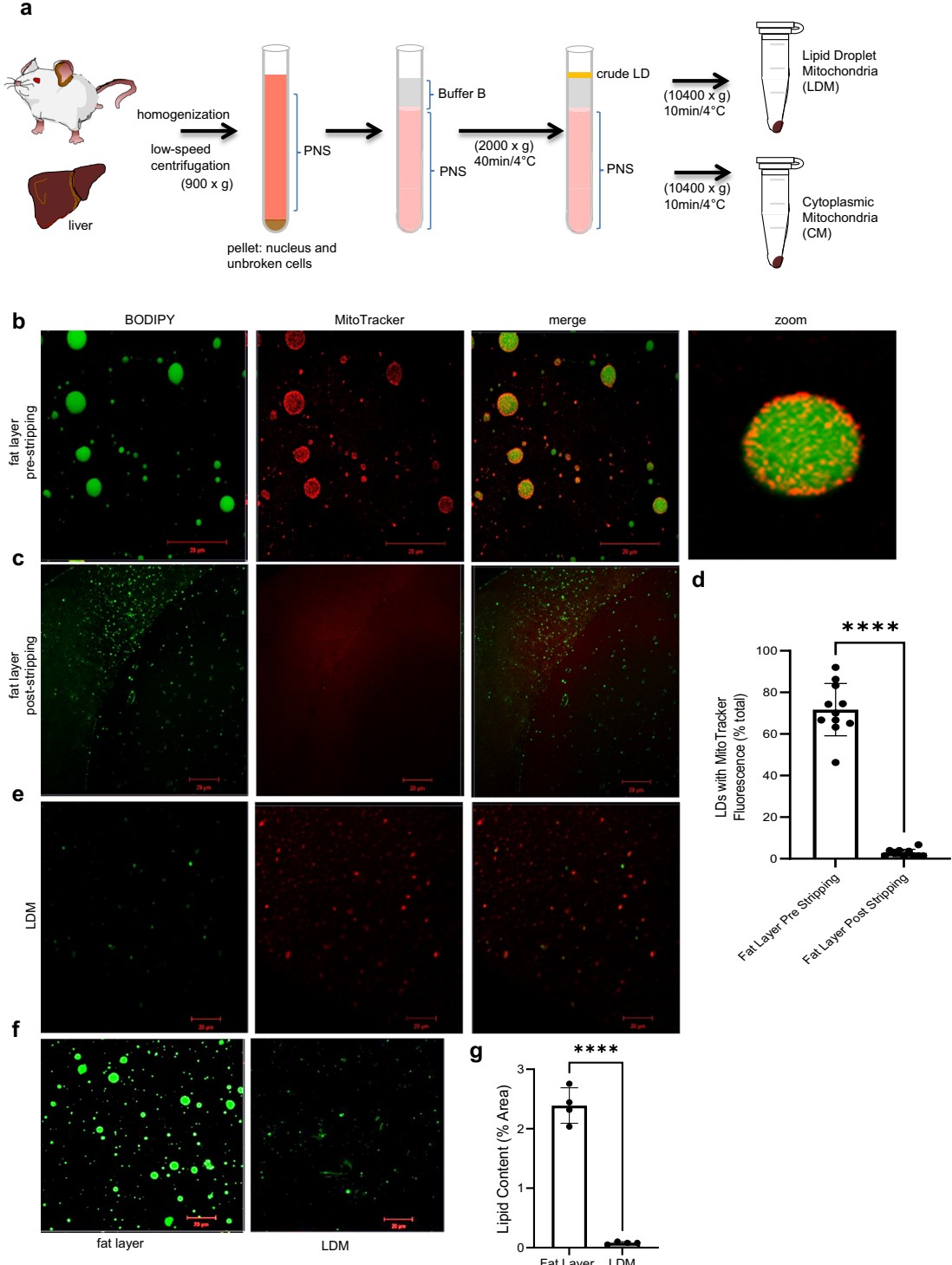

**Fig. 2 | Isolation of Lipid Droplet associated Mitochondria (LDM) by differential centrifugation from rat liver. a** Schematic representation of the method to isolate LDM and Cytosolic Mitochondria (CM) from rat liver. The liver was dissected from Wistar rats and homogenized with a polytron homogenizer. Low-speed centrifugation separated the post-nuclear supernatant (PNS) from the pellet. PNS was layered with buffer B and subjected to centrifugation to separate the fat layer that contains crude LD and the supernatant containing CM. High-speed centrifugation stripped LDM from LD, and CM was pelleted from supernatant PNS fraction. Confocal images of the fat layer before (**b**) and after (**c**) stripping of LDMs. LDs were marked by BODIPY fluorescent dye and mitochondria by MitoTracker red dye. Note the preservation of LD-mitochondria contacts in the fat layer before stripping (b, zoom). Also, note that LDM has been effectively stripped of LD (**c**, MitoTracker) (scale bar, 20 μm). Experiment was performed with two biological replicates and multiple technical replicates were taken from each animal. d) Quantification of the LDs stained with MitoTracker in the fat layer before (**b**) and after stripping of LDMs (**c**) Data from multiple technical replicates from two independent biological replicates were taken and are presented as mean values ± SD. Significance was calculated by two tailed un-paired Student $t$-test. ****$p < 0.0001$. **e** LDM fraction stained with BODIPY and MitoTracker. Note that LDM is primarily free of LDs. **f, g** Comparison of BODIPY staining of the fat layer and LDM (scale bar, 20 μm). Experiment was performed with two biological replicates and numerous technical replicates were taken from each animal. **g** Quantification of the BODIPY intensity in LDs of fat layer (left panel of **f**) and LDMs (right panel of **f**). Data from multiple technical replicates from two independent biological replicates were taken and are presented as mean values ± SD. Significance was calculated by two tailed un-paired Student $t$-test. ****$p < 0.0001$.

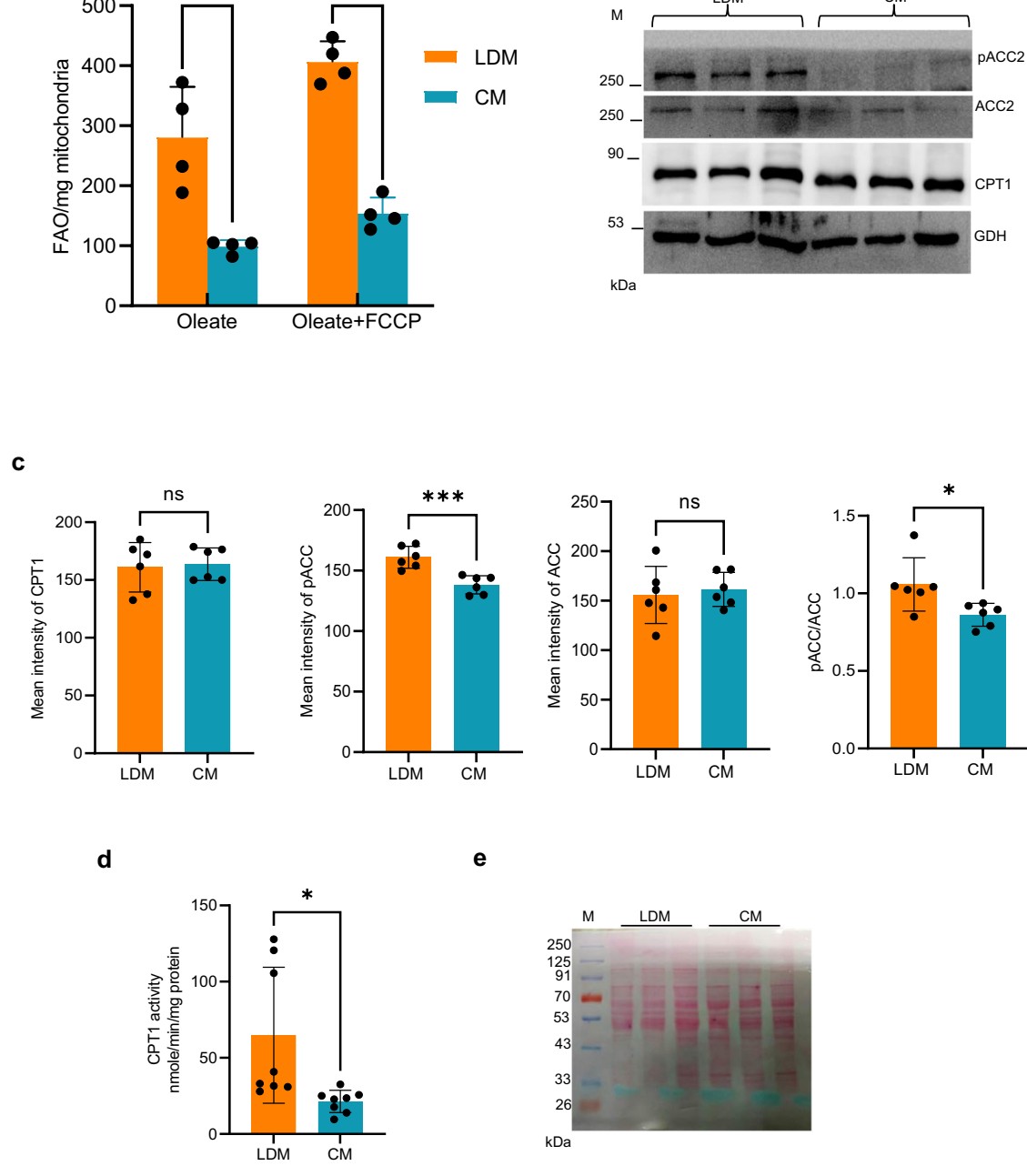

**Fig. 3 | Lipid Droplet associated Mitochondria (LDM) from rat liver is specialized to perform Fatty Acid Oxidation (FAO) during an *ad-libitum* food condition. a–d** Lipid Droplet associated mitochondria (LDM) and Cytoplasmic Mitochondria (CM). **a** LDM has a higher FAO than CM. Quantification of FAO ($n = 4$) in terms of oxygen consumption performed in isolated LDM and CM as described in the Methods section in the presence of oleic acid (OA), and OA + FCCP (maximal respiration induced by chemical un-coupler FCCP). Data are presented as mean values ± SD. Significance was calculated by two tailed un-paired Student *t*-test. **$p = 0.005$, ****$p = 0.00004$. **b** LDM is marked by high pACC2. Shown here is a representative Western blot ($n = 6$) that was probed with antibodies against CPT1, phosphorylated ACC2 (pACC2), ACC2, and glutamate dehydrogenase (GDH) as a mitochondrial loading control. **c** Densitometric analysis of the western blot ($n = 6$) is shown in Fig. 3b for isolated LDM and CM samples from the liver. Data are presented as mean values ± SD. Significance was calculated by two tailed un-paired Student *t*-test. ***$p = 0.0008$, *$p = 0.028$. **d** Quantification of CPT1 activity in LDM and CM ($n = 8$). Data are presented as mean values ± SD. Significance was calculated by two tailed un-paired Student *t*-test. *$p = 0.016$. **e** A Ponceau S stain of the western blot (b) is shown here for total protein loading.

to LDM. In the case of CM, between the succinate and glutamate/malate substrates, succinate appears to be the preferred substrate as there is a relatively higher increase in state III respiration in its presence compared to the glutamate/malate substrate (compare Fig. 4a with 4b). Curiously, succinate and glutamate/malate-induced state III respiration is significantly lower in LDM and hovers close to the corresponding basal state II respiration (Fig. 4a, b). Despite higher FAO in the LDM sample compared to CM, it is surprising that LDM is compromised for coupled respiration and has reduced respiration capacity with reference to TCA substrates (Figs. 3a, 4a, b). The reduced respiration in LDM may be attributed to the presence of uncoupled proteins like UCP2. Hence, we examined the expression of UCP2, the

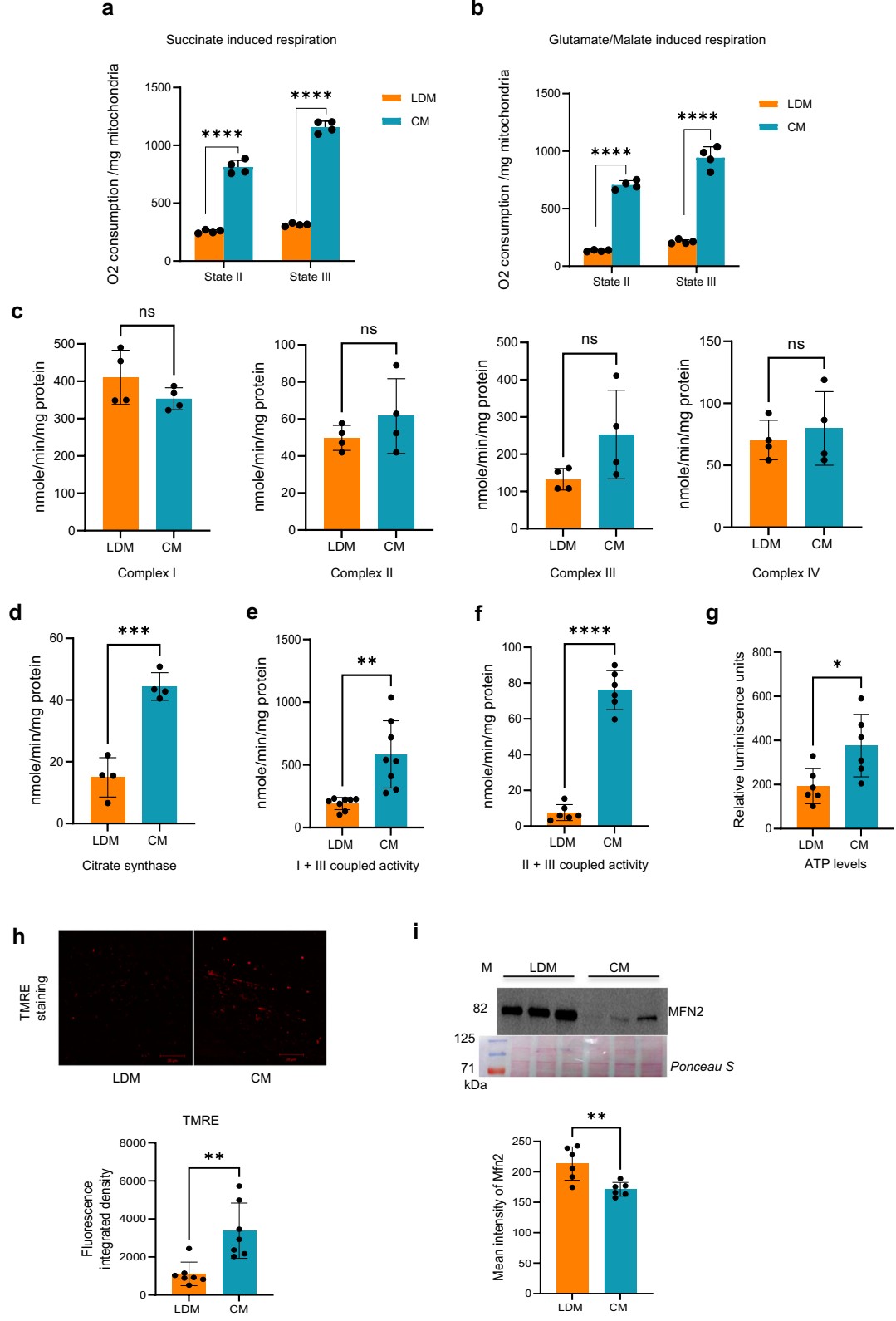

major uncoupler protein in the liver. Interestingly, we find that the expression of UCP2 is nearly abolished in LDM compared to CM, suggesting that uncoupling may not be a reason for the reduced respiration in LDM (Fig. S2a, b).

To further confirm these findings, we measured individual OXPHOS complex activities of the mitochondrial ETC described under the methods. Contrary to our expectations, we find no significant difference in Complex I, II, and IV activities between LDM and CM (Fig. 4c). LDM exhibits an apparent decrease in Complex III activity compared to CM, but it is found to be statistically insignificant (Fig. 4c). Hence, we find the electron flux comparable between LDM and CM. Western blot analysis of OXPHOS complex protein subunits also remains unchanged between LDM and CM except for CIII, which is higher in LDM (Fig. S2a–S2b).

**Fig. 4 | Isolated Lipid Droplet associated Mitochondria (LDM) from the liver exhibit reduced respiration, complex I + III and II + III coupled activities, TCA cycle capacity, ATP levels, and bioenergetics. a–i** LDM and Cytoplasmic Mitochondria (CM) from rat liver. Data are presented as mean values ± SD. Significance was calculated by two tailed un-paired Student t-test. **a**, **b** LDMs are compromised for State II and State III respiration. State II and III respiration rates are driven by succinate (**a**) and glutamate/malate (**b**) in LDM and CM isolated from rat liver. State II respiration quantifies respiration driven by proton leak (no ATP synthesis). State III quantifies respiration driven by ATP synthesis. ($n = 4$) ****$p < 0.0001$. **c** Quantification of the activities of complexes I, II, III, and IV of the Electron Transport Chain (ETC) in LDM and CM ($n = 4$). **d** LDM has reduced TCA flux. Quantification of citrate synthase activity by DTNB absorption assay in isolated LDM and CM ($n = 4$) ***$p = 0.0003$. **e**, **f** LDM trails CM significantly in the channeling of electrons. Super complex I + III and II + III coupled activities were performed using an equal amount of LDM and CM samples. ($n = 6$) **$p = 0.001$, ****$p < 0.0001$. **g** ATP levels diminished in LDM. Quantification of the ATP levels in isolated LDM and CM. ($n = 6$) *$p = 0.02$. **h** CM has a more efficient proton gradient than LDM. Confocal imaging of LDM and CM stained with TMRE, a fluorescent probe that accumulates and stains negatively charged mitochondria (scale bar, 20 μm). Lower panel shows quantification of TMRE fluorescence in LDM and CM. **$p = 0.002$. **i** LDM from rat liver has high MFN2 expression. A representative Western blot probed with MFN2 antibody is shown in the upper panel, and the lower panel is a Ponceau S stain of the Western blot to serve as a loading control. Lower panel shows quantification of the MFN2 protein expression signal from **i**. ($n = 6$) **$p = 0.005$.

Despite comparable electron flux in LDM and CM, LDM manifests poor respiration capacity in the presence of succinate and glutamate/malate (Fig. 4a, b). To address this discrepancy, we reasoned that as the output of the TCA cycle is intimately linked to the ETC, it is plausible that a reduced TCA flux may indirectly affect the respiration capacity in LDM. To validate this hypothesis, we determined the activity of citrate synthase (CS), the first enzyme of the TCA cycle that condenses acetyl CoA and oxaloacetate to form citrate by DTNB absorption assay. The reduction of DTNB is lower in LDM than in CM, suggesting reduced TCA cycle capacity in the former (Fig. 4d). The lag in the pace of oxidation of TCA substrates in LDM may explain the observed decrease in respiration capacity.

Studies have shown that Complex I and Complex III containing super-complex hastens the delivery of electrons via CoQ and thereby increases electron transfer efficiency[26]. To further understand the lowered state III respiration displayed by LDM, we measured super complex I + III and II + III coupled activities described previously[27–29]. Super complex I + III and II + III activities were carried out using equal concentrations of LDM and CM as described in the methods section. Samples were incubated with oxidized cytochrome C, and the super complex I + III activity was initiated by adding NADH. In the case of II + III complex activity, samples were incubated with succinate, and cytochrome C was used to initiate the reaction. We observe a significant six-fold decrease in complex I + III coupled activity in the LDM sample relative to CM, suggesting that the channeling of electrons in LDM is trailing kinetically (Fig. 4e). Similarly, complex II + III coupled activity is also compromised in LDM compared to CM (Fig. 4f). As the flow of electrons and the resultant proton gradient of ETC lead to ATP production, we investigated if the lowered respiration capacity and decreased complex I + III, and complex II + III activities of LDM affect ATP levels. Firefly luciferase luminescence ATP detection assay was performed, and we find decreased ATP levels in the LDM sample compared to CM under an *ad libitum* food condition (Fig. 4g).

Electron flux, electron kinetics, and the resultant proton gradient are critical for ATP generation. Hence, we measured the mitochondrial membrane potential using tetramethylrhodamine ethyl-ester-perchlorate (TMRE). TMRE is a positively charged fluorescent probe that readily accumulates in active, negatively charged mitochondria. An equal amount of LDM and CM populations were stained with TMRE to test their membrane potential at excitation/emission of 549 nm/ 575 nm. LDM exhibits reduced TMRE mean fluorescence compared to CM, which is characteristic of a poor proton gradient (Fig. 4h). The reduced TCA flux with diminished electron kinetics and a poor proton flux derails ATP production in LDM despite LDM displaying higher FAO.

**Liver LDM is associated with higher levels of MFN2**

It is well known that mitochondria go through continuous cycles of fusion and fission to equilibrate the mitochondrial content across the mitochondrial populations of the cell. As LDMs are specialized for FAO, it becomes essential for the liver to secure the integrity of this population from other mitochondrial populations. We hypothesize that LDM may have adopted altered mitochondrial dynamics to maintain this integrity. To test this hypothesis, we checked for the protein levels of mitochondrial fusion protein 2, MFN2, and fission protein, DRP1 and its phosphor form, pDRP1, in rat liver LDM and CM. An equal amount of mitochondria was resolved on SDS-PAGE, western transferred, and probed with anti-MFN2, DRP1, and pDRP1 antibodies. Western blot analysis revealed that MFN2 expression is significantly higher in LDM than in CM (Fig. 4i). Interestingly, previous studies have shown that MFN2 facilitates mitochondria-LD association[7] and is required for cold-induced thermogenesis[30] in brown adipose tissue. Despite a comparable amount of Drp1 in both sets of mitochondria, we find that consistently CM has a relatively higher phosphor-DRP1 presence compared to LDM (Fig. S2d). Overall, LDM exhibits increased mitochondrial fusion dynamics compared to CM in the liver.

To further probe if MFN2 affects the FAO capacity of LDM, we generated lentiviral-mediated *MFN2* shRNA stables in HepG2 cells, as described in the methods section. First, we observed significant silencing of MFN2 by western blot analysis in HepG2 cells transfected with *MFN2* shRNA compared to PLKO.1 vector-transfected control cells (Fig. S3a). Subsequently, we measured the FAO capacity of whole cells and isolated whole mitochondria. We observe a reduction in FAO capacity in whole cells and whole mitochondria when the MFN2 level is decreased (Fig. S3b, S3c). However, we failed to measure FAO in LDM as we could not isolate LDM from HepG2 cells. HepG2 cells did not present any visible fat layer. To circumvent this problem, we treated *MFN2* shRNA and vector control cells with 0.5 mM OA for 48 h to induce the formation of lipid droplets and thereafter isolated crude LDs from the fat layer (Fig. S3d). Fluorescence studies revealed that LD-mitochondria contacts are preserved in the crude LDs obtained after OA treatment of HepG2 cells, which includes control cells and *MFN2* shRNA transfected cells (Fig. S3e). However, the size of the LDs appears to be larger in HepG2 cells transfected with *MFN2* shRNA compared to control cells (Fig. S3e). Though we could isolate LDM and CM from MFN2-depleted HepG2 cells, we could not isolate a sufficient amount from control cells for further characterization. *MFN2* silencing did not alter the protein expression of OXPHOS complex subunits, nor did OA treatment affect MFN2 expression in total cell lysates (Fig. S3d and S3f).

We observe an apparent increase in LD size when MFN2 is depleted in HepG2 cells upon OA treatment (Fig. S3e). To ascertain if there is any correlation between FAO and LD size, we examined the size of LDs in HepG2 vector control, and *MFN2* silenced cells followed by staining of LDs with the neutral lipid dye BODIPY as described in the methods. Lack of sufficient MFN2 apparently increases the size of LDs even in the absence of OA (Fig. S3g). To ensure that the observed rise in LD size during MFN2 depletion is not a result of increased LD biogenesis or increased translocation of FA to LDM, we checked for LD marker protein, perilipin 2 (PLN2), and CPT1 protein levels. The levels of PLN2 and CPT1 in *MFN2* silenced cells were comparable to control cells (Fig. S3h). Our results show that the depletion of MFN2 affects FAO and LD size in HepG2 cells. However, further studies are required to ascertain the precise role of MFN2.

## HFD induced NAFLD diminishes FAO capacity and alters respiration in LDM

To date, the exact role of LDMs in NAFLD condition is unknown. Significantly, impaired and enhanced mitochondrial oxidative functions have been implicated in NAFLD[31]. Our method to isolate LDM from the liver provides a handle to determine the role of LDMs in NAFLD condition. Towards this, we isolated LDM from the standard diet (SD) fed and high-fat diet (HFD) induced NAFLD animals. At the outset, weight-matched Wistar rats were divided into two groups, SD and HFD. HFD was fed to rats ($n = 10$) for 16 weeks to induce NAFLD condition, while SD-fed rats ($n = 10$) were used as controls (Fig. S4). At the end of 16 weeks, there was no apparent weight difference between the groups despite HFD inducing a gain in body weight between 7-8 weeks (Fig. S4c). However, the livers in HFD-fed animals had gained more weight significantly, thus increasing the liver/body weight ratio (Fig. S4e). Serum FA examination revealed that animals in the HFD group had elevated serum triglyceride and total serum cholesterol levels (Fig. S4g and S4h). Hematoxylin-eosin (H&E) staining of liver sections from the HFD group presented massive macro and micro steatosis (Fig. S4f). Gross liver morphology examination highlighted NAFLD characteristics in the HFD group, while the SD group lacked them (Fig. S4a).

After establishing a NAFLD animal model system, we isolated LDM from SD and HFD animal groups as described in the methods section. We observe HFD-fed animals harboring higher levels of LDM. We initially examined their FAO capacity to determine the functional relevance of LDMs. HFD induces a significant reduction in FAO capacity in LDM that is lower than CM from the same group and LDM and CM from the SD group (Fig. 5a). The lowered FAO may be attributed to the marked steatosis and LD accumulation that characterizes NAFLD (Fig. S4a). HFD CM had slightly better FAO than LDM, suggesting that HFD may have caused CM to adapt to the high-fat diet (Fig. 5a). Interestingly, OA-induced FAO capacity increased in SD LDM compared to SD CM underscoring the specialized function of LDM in the liver, as shown earlier (Figs. 3a and 5a). This is further reinforced when we look at pACC2 and ACC2 levels. LDMs are enriched in pACC2 and ACC2 compared to CM in the SD group; however, they are dramatically reduced in NAFLD condition (Fig. 5e).

Many studies have shown NAFLD to be correlated with higher mitochondrial respiration capacity[31,32]. Hence, we examined state III respiration in LDM isolated from HFD-induced NAFLD animals in the presence of succinate or glutamate/malate (Fig. 5b, d). HFD treatment elevates the respiration capacity of both LDM and CM compared to their counterpart in SD conditions in the presence of both succinate and glutamate/malate. However, HFD CM exhibits a relatively higher increase than HFD LDM (Fig. 5c, d). The relatively greater increase in state III respiration in the presence of malate/glutamate in both HFD LDM and HFD CM compared to SD and *ad libitum* food condition is intriguing (Figs. 4b and 5b–d).

## Discussion

As a metabolic hub, the liver is empowered with various functions that include regulating the homeostasis of glucose, fatty acids, and amino acids, besides detoxification of undesirable metabolites, glycogen storage, and alcohol metabolism. The importance of liver functions is underscored by its rejuvenation capacity. Any perturbations in liver functions can lead to numerous disorders, including FLD and cancer. The liver is critically dependent on the mitochondria for its energy requirement in the form of ATP so that all its functions can operate optimally. Besides numerous other functions, mitochondria host three crucial metabolic pathways, the TCA cycle, FAO, and the ETC, that together orchestrate substrate utilization, energy generation, and redox homeostasis. While the TCA impinges on ETC, FAO impinges on both TCA and ETC for oxygen consumption and energy production. Impairment in any of the mitochondrial functions can have deleterious

consequences, and in the context of the liver, it can lead to hepatic disorders[21].

Despite the importance of mitochondria in liver functions, little progress has been made in unraveling the role of mitochondria in the liver. Limited evidence describes that mitochondria in brown adipose tissue are categorized into metabolically distinct sub-populations to perform two antagonist functions simultaneously: fatty acid oxidation and LD biogenesis[8,31,33]. This finding raises more questions: can different sub-populations of mitochondria co-exist within the same cell? Is the existence of discrete populations of mitochondria a normal phenomenon? Or is it an adaptation to a physiological demand or a pathogenic condition? Indeed, the liver is specialized as it synthesizes lipids under prolonged fasting and pathological conditions such as obesity and NAFLD[34,35]. Hence, it is essential to examine the liver mitochondria and understand their adaptation to the lipid environment in the liver. The lack of a method to isolate mitochondrial sub-populations from the liver has severely curtailed our understanding of mitochondrial functions in the liver and has fueled contradictory theories about the cause of NAFLD[31]. Historically, multiple attempts were made to isolate LD-enriched buoyant fractions that used high-speed ultracentrifugation[22,36–39]. LDs are fragile, and they fall apart during the ultracentrifugation step. In 2018, Benador *et al.* employed low-speed centrifugation and successfully isolated LD-associated mitochondria from BAT. Unlike BAT, that is very rich in fat; low-speed centrifugation does not yield a clear fat layer from the liver. Hence, we made a few changes in the LDM isolation procedure described in the methods and shown in Fig. 2. Importantly, we show that the liver harbors two kinds of mitochondria, CM and LDM (Figs. 1 and 2). The latter specialized for FAO to cope with the increased fat present in the liver under unrestricted nutrient-rich conditions. Interestingly, we find mild lysosome and significant ER presence in the LDM sample. Further studies are required to probe inter-organellar communication in the context of regulating LDM functions.

A recent study has shown that mitochondria associated with LDs in BAT exhibit increased pyruvate oxidation, electron transport, and ATP synthesis besides supporting LD biogenesis. Curiously, the mitochondria are shown to depart from LDs when β-oxidation is activated in BAT[8]. In contrast to these observations in BAT, in the case of the liver, we find that LDM is associated with higher FAO under an *ad libitum* food condition (Fig. 3).

Between TCA and FAO, the ratio of NADH/FADH2 is higher for the former; hence, the ratio of ATP/oxygen consumption is also higher. FAO leads to more ATP; however, more oxygen is consumed per mole of ATP produced. It is well known that decreasing the proton gradient by uncoupling propels FADH2 oxidation. In the liver, though we see increased FAO in LDM (Fig. 3), it is not associated with higher ATP production (Fig. 4g). Based on the evidence presented here, LDM displays robust FAO correlated with high oxygen consumption; however, TCA shows a deficiency in TCA flux (Fig. 4d) and is associated with poor state II and state III respiration (Fig. 4a, b). Intriguingly, the impairment in complex I + III and II + III electron channeling (Fig. 4e, f) is apparently exclusively affecting TCA-driven oxygen consumption and not the FAO-dependent ETC pathway (Fig. 4c). Though individual OXPHOS complex activities are comparable in CM and LDM, super complex I + III and II + III activities are decreased in LDM. We attribute this difference to the low endogenous pool of CoQ in LDM and super complex respiring pools[40–42]. Our TMRE staining results also support that LDM is less active than CM, as evidenced by a reduced mitochondrial membrane potential at LDM (Fig. 4h). As the energy produced from FA is far greater than from glucose, derailment in ETC on FAO-linked oxygen utilization might be masked by a hyperactive FAO pathway.

The human cell, particularly the mitochondria, has evolved a system of checks and balances for the continuous and intricate orchestration of its many complex processes that require the export

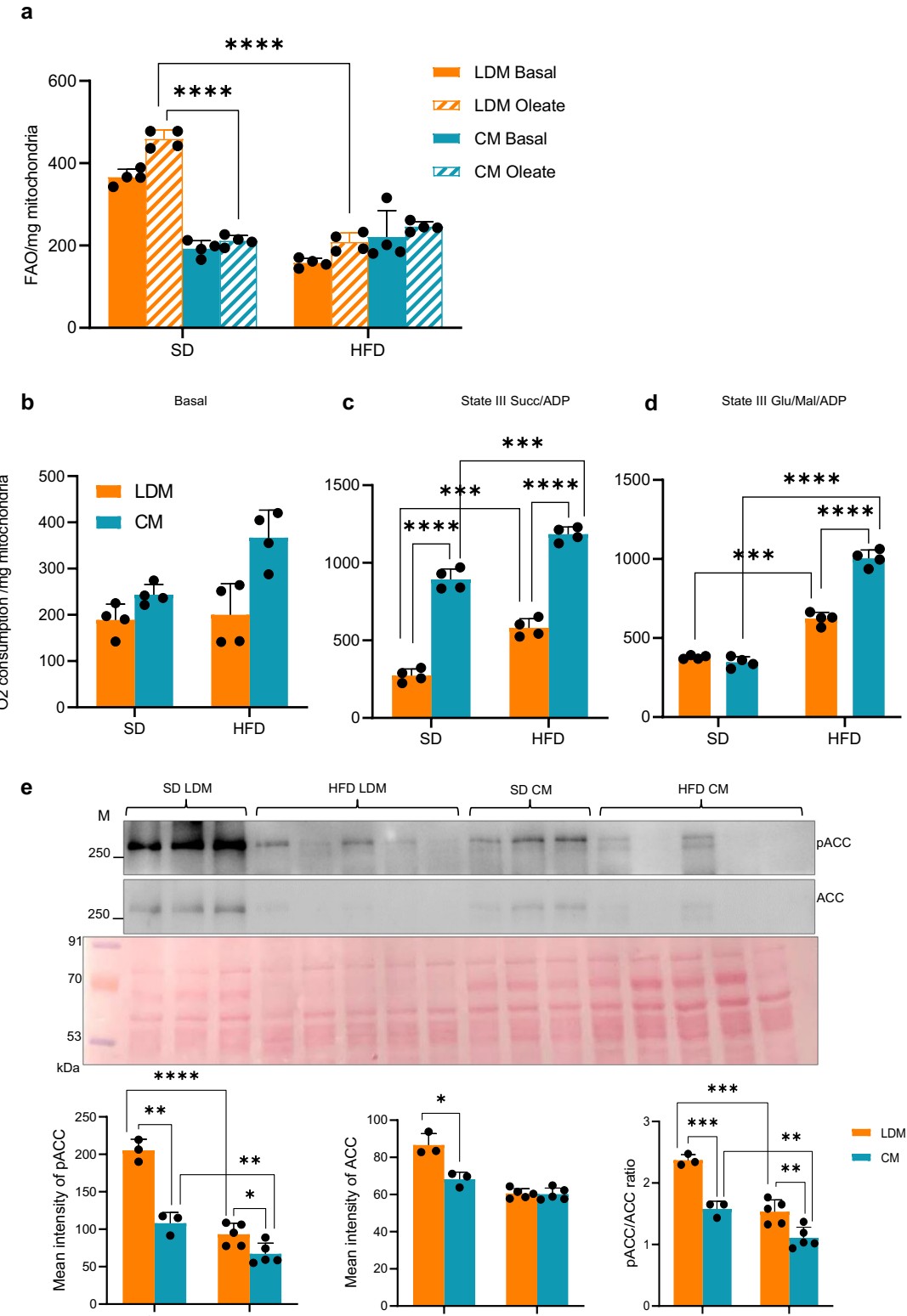

**Fig. 5 | Fatty Acid Oxidation (FAO) is significantly decreased in liver Lipid Droplet associated Mitochondria (LDM) isolated from High Fat Diet (HFD) induced Non-Alcoholic Fatty Liver Disease (NAFLD) model rat animals. a** Wistar rats were fed with Standard Diet (SD) to serve as control or with HFD to induce NAFLD, as described in the Methods section and as shown in Supplementary Fig. 4. Cytoplasmic Mitochondria (CM) and LDM were isolated from SD and HFD-fed animals. FAO was monitored as the oxygen consumption rate in the presence of Oleate or the absence (Basal) of Oleic acid. (*n* = 4). Data are presented as mean values ± SD. Significance was calculated by two tailed un-paired Student *t*-test. ****p* < 0.0001. **b–d** Respiration capacity in LDM and CM isolated from control SD-

fed and NAFLD induced HFD-fed model animals. Basal respiration in the absence of any substrate and State III respiration in the presence of Succinate+ADP or Glutamate/Malate+ADP were monitored in CM and LDM samples isolated from control SD-fed and NAFLD induced HFD-fed model animals. (*n* = 4). Data are presented as mean values ± SD. Significance was calculated by two tailed un-paired Student *t*-test. ****P* < 0.0001, ***P* < 0.001. **e** Total and phosphor ACC levels are significantly decreased in LDM and CM of NAFLD. Western blot (*n* = 3 for SD and *n* = 5 for HFD) probed with antibodies against phosphorylated ACC2 (pACC2) and ACC2. Data are presented as mean values +/− SD. Significance was calculated by two tailed un-paired Student *t*-test. ****P* < 0.0001, ***P* < 0.001, ***P* < 0.01 and **P* < 0.05.

and import of metabolites, proteins, and cofactors. The delicate balance between fatty acid synthesis and β-oxidation is one example where a repertoire of protein factors function as gatekeepers, including CPT1, ACC, and malonyl CoA decarboxylase (MCD)[43]. While CPT1 is exclusively present in the mitochondria, both ACC and MCD exist in cytosol and mitochondria. Malonyl CoA serves as the chain elongating unit for fatty acid synthesis and represses fatty acid oxidation. CPT1, the rate-limiting enzyme of fatty acid oxidation, is allosterically inhibited by malonyl CoA[24]. However, malonyl CoA levels are kept in check by the dual action of ACC and MCD. ACC increases malonyl CoA by converting acetyl CoA to malonyl CoA, while MCD acts on malonyl CoA and creates acetyl CoA. Our results show that though CPT1 protein levels are not altered in LDM (Fig. 3b, c), we find increased activity of CPT1 (Fig. 3d) and phosphorylation of ACC2 (Fig. 3b, c). Interestingly, it has been shown that mice lacking ACC2 have enhanced fatty acid oxidation, decreased LDs, and reduced triglyceride content in the liver[25]. Taken together, ACC2 inactivation and increased CPT1 activity are couple of the mechanisms that LDM adopts to regulate LD homeostasis under a surplus nutrient condition.

MFN2, the mitofusion protein 2, plays a central role in mitochondrial fusion with other fusion and fission proteins to maintain the integrity of mitochondria. The level of MFN2 varies depending on the tissue and is predominantly present in the brain. Deficiency of MFN2 has been associated with various myopathies, including obesity, diabetes, cancer, Alzheimer's, and Parkinson's diseases[44–47]. Besides playing a role in mitochondrial fusion, MFN2 regulates mitochondria's tethering to ER and microtubules[48,49]. Incidentally, MFN2-perilipin1 association facilitates mitochondria-LD linkage in response to stimulants, and thereby MFN2 increases lipolysis and affects energy homeostasis in BAT[7]. It has been shown that peridroplet mitochondria (PDM) in BAT are associated with higher levels of MFN2. Intriguingly, PDM-associated MFN2 does not contribute to lipid expansion nor affects FAO in BAT[8]. Nevertheless, our results show that, akin to PDM in BAT, LDM in the liver is associated with significantly higher levels of MFN2 compared to CM (Fig. 4i).

In this study, we have isolated two populations of mitochondria from HFD-induced NAFLD animals that exhibit fatty liver, triglyceride accumulation, and LD expansion. Despite no change in body weight, the livers in HFD-fed animals show an increase in weight. FAO capacity is decreased in NAFLD LDM compared to SD LDM, while there is no significant variation in the CM population from both animal groups (Fig. 5a). Our studies show that during NAFLD, in the presence of malate/glutamate, there is a metabolic adaptation to increase energy efficiency by favoring glucose oxidation to lipid oxidation by both CM and LDM. This adaptation is illustrated by increased state III respiration in NAFLD condition (Fig. 5c, d). Interestingly, it has been suggested that lipid homeostasis is dictated more by mitochondrial substrate preference than energy[31]. The increased LD biogenesis and expansion observed in NAFLD could be due to decreased FAO in the LDM population and an adaptive increase in state III respiration.

NAFLD, whose prevalence is increasing worldwide and affects more than 25% of the population. As it is a major risk factor for non-alcoholic steatohepatitis (NASH), cirrhosis, and liver cancer, understanding the molecular mechanisms of liver function, especially different mitochondrial populations, is critical for understanding the disease and developing therapies for this unmet medical condition. Based on the evidence presented, we isolated two separate and segregated populations of mitochondria from the liver and showed that the LDM is specialized and adapted for FAO. We demonstrated that the hallmarks of liver LDM under an *ad libitum* condition include increased FAO, decreased TCA flux, decreased membrane potential, decreased complexes I + III and II + III activities, decreased ATP levels, increased CPT1 activity, elevated MFN2 levels, inactivated ACC2, and decreased LD size. Interestingly, several studies have shown reduced *MFN2* expression in humans suffering from NASH and in mouse NASH

models. Most importantly, restoring the expression of *MFN2* ameliorates the disease[50]. These results highlight the importance of LDM in the liver and its role in FAO in maintaining the liver lipid metabolism. This study provides another dimension in ameliorating NAFLD and opens up new avenues for the therapeutic interventions of fatty liver disease.

## Methods

### Ethics
All experiments were performed according to the Indian Institutional Ethical Committee (IEC) Board guidelines and approved by the Institutional Animal Ethics Committee (IAEC) of the University of Hyderabad (UH/IAEC/NBVS/2021-22/12), Hyderabad, India.

### Cell culture
HEK293T cell line was purchased from the American Type Culture Collection (ATCC), USA. HepG2 cell line was purchased from the cell line repository of the National Centre for Cell Science (NCCS), India.

HEK293 T and HepG2 cells were cultured in Dulbecco's modified Eagle's medium (DMEM) supplemented with 10% (v/v) fetal bovine serum, 100 U/ mL penicillin/streptomycin (Invitrogen, Carlsbad, CA, USA) at 37 °C and under 5% $CO_2$. The culture medium was changed every other day, and the cells sub-cultured once they attained 70–80% confluence.

### Lentiviral vector production
Lentiviral vectors with PLKO.1 and *MFN2* shRNA were produced in HEK293T cells. HEK293T cells were cultured in DMEM containing 10% (v/v) fetal calf bovine serum at 37 °C under an atmosphere of 5% $CO_2$. For stable knockdown of *MFN2* in HepG2 cells, independent small hairpin RNA (shRNA) targeting *MFN2* (TRCN0000082683; Sigma Aldrich) was used, and PLKO.1 empty vector (SHC001; Sigma Aldrich) was used as a control. Cells were grown in 60 mm flasks transfected with lentiviral packaging plasmids (VSVG – 1.625 μg; ΔR – 0.625 μg; Rev – 0.875 μg) and carrier plasmids (PLKO.1 – 2.5 μg; *MFN2* shRNA – 2.5 μg) by using lipofectamine transfection reagent (Invitrogen). After ten hours of transfection, the medium was changed, and supernatants containing the virus were collected every 24 h for two days. Supernatants were stored in aliquots at −80 °C until further use.

### Generation of HepG2 cells stably expressing *MFN2* shRNA
For the transduction of HepG2 cells, 3 ×10⁵ cells were seeded in 60 mm flasks with the appropriate amount of viral supernatants and 5 μg/mL polybrene. Forty-eight hours post-transduction, cells were subjected to 4 μg/mL Puromycin selection. Cells were analyzed for *MFN2* silencing by immunoblotting with MFN2 antibody.

### Animals
Three-month-old male Wistar rats (*n* = 6) were used for all control experiments. Six-week-old animals were fed a standard diet and a high-fat diet. All animals were maintained in a pathogen free facility at room temperature 23 ± 1 °C in 12:12 h light and dark cycles. Animals were euthanized by cervical dislocation after anaesthetization with isoflurane. All experimental procedures were performed according to the Indian Institutional Ethical Committee (IEC) Board guidelines, and animals were given *ad libitum* food and water.

### Isolation of LDM from rat liver
The liver from Wistar rat (*n* = 6) was harvested and washed with 1 X phosphate-buffered saline (PBS) to remove blood contamination. The liver was weighed, minced, and suspended in Sucrose-HEPES-EGTA (SHE) buffer supplemented with 2% BSA (250 mM Sucrose, 5 mM HEPES, 2 mM EGTA, 2% fatty acid-free BSA, pH 7.2). The tissue suspension was mechanically homogenized using a

polytron homogenizer (3 sec X 4 pulses and 15 rpm). The homogenate was transferred into a 50 mL falcon tube (Corning) and centrifuged in a swinging bucket rotor at 900 X g for 10 min at 4°C. The post-nuclear supernatant (PNS) was collected and carefully layered with Buffer B (20 mM HEPES, 100 mM KCl, 2 mM MgCl$_2$, pH 7.4) without disturbing the below layer. Next, centrifugation was carried out in a swinging bucket rotor at 2000 X g for 40 min at 4°C to allow the fat layer to separate. The fat layer that contains crude LD appears as a translucent band on the top of the gradient. The fat layer was carefully collected and re-suspended in SHE buffer without BSA. This fat layer suspension was centrifuged at 10400Xg for 10 min at 4°C in a fixed rotor. We applied mild vortex prior to this step to break the LD-mitochondria contacts. The pellet obtained after centrifugation contains LDM, and it was re-suspended in mitochondrial resuspension buffer (MRB; 250 mM Mannitol, 5 mM HEPES pH 7.4 and 0.5 mM EGTA). The cytosolic fraction beneath the fat layer was subjected to centrifugation at 10400Xg/10 min/4°C in a fixed rotor. The pellet obtained after centrifugation contains CM and was re-suspended in MRB.

### Electron microscopy

For Transmission Electron Microscopy (TEM), animals were perfused *via* the vascular system by injecting 4% paraformaldehyde and 1% glutaraldehyde through the heart's left ventricle with the help of a peristaltic pump under standard pressure conditions. Liver from Wistar rat (*n* = 3) was harvested, cut into 2 mm fragments, and thereafter fixed in 2.5% glutaraldehyde and 4% paraformaldehyde in 0.1 M PBS, pH 7.4 at room temperature for 24 h. This was followed by treatment with osmium tetroxide for 3 h and dehydrated in a series of graded alcohol, infiltrated and embedded in Araldite resin, and allowed for polymerization at 80 °C for 72 h. Ultrathin (60 nm) sections were made in Leica Ultra cut UCT-GA-D/E-1/00, mounted on copper grids, and stained with Uranyl acetate. Imaging was performed on the JEOLJEM2100 TEM.

### Fluorescence microscopy

Crude LD from HepG2 cells, and crude LD and LDM from rat liver were isolated and stained with 1 mM MitoTracker red 579/599 nm, 1 mM BODIPY 493/503 and placed on a 1 mm glass slide and covered with cover glass. Fluorescence was measured, and image analysis was performed using Image J software. LDM and CM were isolated from adult rat liver, stained with TMRE (20 nM), and fluorescence was measured. Fluorescence was quantified using Image J. Post-treatment with or without OA (0.5 mM) PLKO.1 and Mfn2 silenced cells were washed with 1X PBS and fixed in methanol for 20 min at −20 °C. Post-fixation cells were washed twice with 1X PBS and stained with BODIPY (1 μg/mL PBS) for 30 min at room temperature. Remnants of stain were removed by washing once with 1X PBS and mounted with DAPI mounting medium (Abcam). Imaging was performed using a laser scanning confocal microscope (Model: NLO 710, CARL ZEISS). Fluorescence was measured and quantified using Image J software.

### Immunoblotting

The Lowry method quantified mitochondrial protein content according to the manufacturer's instructions (Biorad). Equal concentrations of mitochondrial protein from LDM and CM preparations were resolved on SDS-PAGE and western transferred to a nitrocellulose membrane. Blots were blocked with 5% non-fat milk in TBST for 1 h, followed by incubation with primary antibodies overnight at 4 °C. The next day, blots were washed with TBST and probed with HRP-tagged secondary antibodies. Details of primary and secondary antibodies used in this study are as follows: Mitofusin2 (D2D10) Rabbit mAb Catalogue No. 9482 (Cell signaling Technologies), dilution 1:1000; ACC (C83B10) Rabbit mAb Catalogue No. 3676 (Cell signaling Technologies), dilution 1:1000; Phosphor-ACC (Ser79) (D7D11) Rabbit mAb Catalogue No. 11818 (Cell signaling Technologies), dilution 1:1000; Anti-CPT1A (8F6AE9) mouse mAb Catalogue No. ab128568 (Abcam), dilution 1:2000; Total OXPHOS rodent WB antibody cocktail, Catalogue No. ab110413 (Abcam), dilution 1:2000; Anti-ADFP (Perilipin2) (EPR3713) Rabbit mAb Catalogue No. ab108323 (Abcam), dilution 1:2000; Anti-β-actin Catalogue No. A3854 (Sigma Aldrich), dilution 1:10000; DRP1 rabbit Ab, Catalogue No. 8570 (Cell signaling Technologies), dilution 1:1000; Phosphor-DRP1 (Ser616) Rabbit Ab, Catalogue No. 3455 (Cell signaling Technologies), dilution 1:1000; Calreticulin, Rabbit polyclonal Ab, Catalogue No. ab2907 (Abcam), dilution 1:2000; Lamp1, Rabbit polyclonal Ab, Catalogue No. ab62562 (Abcam), dilution 1:2000; GRP78, Rabbit polyclonal Ab, Catalogue No. ab21685 (Abcam), dilution 1:2000; GAPDH, mouse monoclonal Ab, Catalogue No. ab8245 (Abcam), dilution 1:2000; UCP2, rabbit monoclonal Ab, Catalogue No. 89326 (Cell signaling Technologies), dilution 1:1000; GDH, Rabbit Ab, Catalogue No. NB600-853 (Novus Biologicals), dilution 1:2000; Secondary Antibody: Peroxidase AffiniPure Goat Anti-Rabbit IgG (H + L), Cat No. 111-035-144 dilution 1:10000, and Peroxidase Affiniy Pure Goat Anti-Mouse IgG (H + L), Cat No. 115-035-146 dilution 1:10000 (Jackson Immuno Research Laboratories); Blots were developed using ECL reagents, and imaging was performed with the help of a BIORAD analyzer. Densitometry analysis was performed by using Image J software.

### ATP assay

This assay was performed using a Luminescent ATP detection assay kit from Abcam (ab113849). An equal amount of isolated mitochondria (LDM and CM) were suspended in 20 mM Tris pH 7.5 buffer. 100 μL of this mixture was combined with 50 μL of detergent and 50 μL of luciferase-luciferin enzyme-substrate combination in an ELISA standard plate. The plate was placed on an orbital shaker for 5 minutes, and luminescence was recorded immediately thereafter.

### Mitochondrial electron transport chain complex activities

All mitochondrial electron transport chain complex activities were performed according to a previously published protocol[27–29].

**Coupled activity I + III**. An equal amount of isolated mitochondria was suspended in 0.05 M KP (potassium phosphate) buffer containing 300 μM sodium azide, BSA (1 mg/mL), and oxidized cytochrome C (50 μM). The reaction was started by supplying an electron donor, NADH (200 μM), to the mitochondria. The coupled activity was measured by recording the increase in absorbance spectrophotometrically at 550 nM.

**Coupled activity II + III**. An equal amount of isolated mitochondria was suspended in 0.02 M KP buffer containing 300 μM sodium azide, BSA (1 mg/mL), and succinate (10 mM). The mitochondrial suspension was incubated at 37°C for 10 min. The reaction was started by supplying oxidized cytochrome C (50 μM) to the mitochondria. The coupled activity was measured by recording the increase in absorbance spectrophotometrically at 550 nM.

**Complex I activity**. An equal amount of isolated mitochondria was suspended in 0.05 M KP buffer containing 300 μM sodium azide and BSA (3 mg/mL). Immediately, an electron donor, NADH (100 μM), was added to the suspension. The reaction was initiated by adding an electron acceptor, ubiquinone (60 μM), to the mitochondria. Complex I activity was measured as the rate of NADH oxidized to NAD + that correlates directly to a decrease in absorbance at 340 nM in a spectrophotometer.

**Complex II activity**. An equal amount of isolated mitochondria were suspended in 0.05 M KP buffer containing 20 mM succinate, 300 μM

sodium azide, 80 μM DCPIP, and BSA (1 mg/mL). The mitochondrial suspension was incubated at 37⁰C for 10 min. Complex II activity was initiated with the addition of decyl ubiquinone (50 μM), and its activity was monitored in a spectrophotometer as a decrease in absorbance at 600 nM.

**Complex III activity.** An equal amount of isolated mitochondria were suspended in 0.05 M KP buffer containing 100 μM EDTA, 75 μM cytochrome C, and 300 μM sodium azide. After adding decylubiquinol (DubH2; 100 μM), Complex III activity was measured. Decyl ubiquinol transfers electrons to cytochrome C and reduces it. The reduction of cytochrome C can be observed by monitoring its absorbance at 550 nm in a spectrophotometer.

**Complex IV activity.** An equal amount of isolated mitochondria was added to the 0.05 M KP buffer containing 50 μM reduced cytochrome C. The decrease in absorbance at 550 nM correlates to the direct oxidation of cytochrome C by complex IV.

### Carnitine-palmitoyl transferase-1 assay
Carnitine-Palmitoyl transferase-1 (CPT1) enzymatic activity was measured as described previously[51]. This assay measures the initial rates of CoA-SH formation by DTNB reagent in the presence of palmitoyl CoA and mitochondria. The reaction mixture contains 116 mM Tris-HCl pH 8.0, 0.09% Triton X 100, 1.1 mM EDTA, 0.12 mM DTNB, 0.035 mM palmitoyl CoA (Sigma), and 1mM L-Carnitine or 11 mM D-Carnitine. The reaction was started by adding equal concentrations of LDM and CM, and the increase in absorbance was measured for four minutes. The CPT1 activity was calculated by subtracting the amount of CoA-SH formed in the presence of D-Carnitine from the amount of CoA-SH formed in the presence of L-carnitine spectrophotometrically at 412 nM.

### Fatty Acid Oxidation (FAO)
FAO was determined using a fatty acid complete oxidation assay kit from Abcam (ab222944). The assay was performed according to the manufacturer's protocol. This assay uses FAO substrate, Oleate conjugated with BSA, and FAO modulator FCCP. In the presence or the absence of FAO substrate, Oleic acid, FAO can be determined. An equal number of cells or an equal amount of isolated mitochondria were incubated in a fatty acid measurement medium that contained 150 μM Oleate conjugated with BSA and 0.5mM L-carnitine in a standard ELISA plate. Extracellular oxygen reagent was added to the mitochondria before sealing each well with highly sensitive mineral oil to limit the back diffusion of oxygen. This assay was designed based on the principle that molecular oxygen quenches the oxygen consumption reagent. Therefore, the increase in fatty acid-driven oxygen consumption is monitored for one hour as an increase in fluorescence at oxygen absorption spectra (Excitation and Emission at 380-650 nM). Basal FAO in mitochondria was measured in the presence of carnitine without Oleate.

### Mitochondrial respiration
Mitochondrial respiration was determined using Abcam's extracellular oxygen consumption assay kit (ab197243). The required concentration of isolated mitochondria (0.5-1.5 mg/mL) was diluted in mitochondria measurement buffer (MMB) that contains 250 mM sucrose, 15 mM KCl, 1 mM EGTA, 5 mM MgCl₂, 30 mM K₂HPO₄, pH 7.4. State II respiration was tested by supplying substrates such as glutamate/malate (12.5 mM, final concentration) and succinate (25 mM, final concentration). State III respiration was tested by supplying ADP (1.65 mM, final concentration) to the isolated mitochondria in the presence of substrates. The fluorescence was measured at oxygen absorption spectra (Excitation and Emission at 380–650 nM).

### Citrate synthase assay
Citrate Synthase activity was performed according to Spinazzi et al., 2012, by using DTNB. Isolated mitochondria were incubated with 100 mM Tris buffer containing 0.1% triton x 100, 100 μM DTNB, and 0.3 mM Acetyl Co-A for 2 min. The reaction was initiated by adding 0.5 mM oxaloacetic acid (OAA). The increase in absorption at 412 nm is directly proportional to the citrate synthase activity.

### Experimental design and diet for inducing NAFLD
Male Wistar rats at 4 weeks of age (~60 g) were allowed to acclimatize in the animal housing facility for 15 days before experimentation. During acclimatization, rats were pair housed and provided standard rodent chow and water under an *ad libitum* condition. The animals were maintained at ambient temperature and under 12 h light and 12 h dark cycles. Following acclimatization, at 6 weeks of age, the rats were randomly assigned to one of two groups (n = 10/group). Group 1 animals were given *ad libitum* food of standard diet and water. Group 2 animals were provided *ad libitum* food of high sucrose and high-fat diet (HFD) supplied by the National Institute of Nutrition (NIN, Hyderabad, INDIA) for 16 weeks. Bodyweight was monitored before, during, and after the experiment. The detailed composition of standard and HFD is shown in Supplementary section.

### Sample preparation, histological examination, serum triglyceride, and total serum cholesterol determination
Blood samples were collected and submitted to a pathology laboratory for total serum triglyceride and cholesterol analysis. Analysis was done using Aspen Chem Ultra fully automatic biochemistry analyser. A liver fragment from all rats was fixed in 4 % paraformaldehyde in 0.1 M phosphate buffer (PB) (pH 7.4) for 2–4 h. Paraformaldehyde fixed paraffin-embedded tissue sections were used for histochemical analysis. 3 μ thick sections were obtained using a microtome and stained with hematoxylin and eosin (H&E). The slides were examined under an optic microscope.

### Statistical analysis
All the results are expressed as mean± SD. Statistical analysis was done by Unpaired Student's *t*-test. A value of $P \leq 0.05$ was considered significant. GraphPad Prism 9.4.1 was used for all statistical analyses and plotting graphs.

### Reporting summary
Further information on research design is available in the Nature Portfolio Reporting Summary linked to this article.

## Data availability
The data supporting the findings of this work are available within the paper and in the Supplementary Information files. Source data are provided as a source data file. Source data are provided with this paper.

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

## Acknowledgements

The authors acknowledge all the members of the NBVS lab for their input. This work was supported by a grant from the Institute of Eminence-University of Hyderabad (UoH-IOE-RC3–21-010) to N.B.V.S. and DBT-BUILDER (BT/INF/22/sp41176/2020) grant to School of Life Sciences. N.K.T. thanks the Council of Scientific and Industrial Research (CSIR), India, for a Research Associate Fellowship (File No. 09/414(1192)/2019-EMR-I), and U.M. is a recipient of the DBT Women BioCARe grant.

## Author contributions

N.B.V.S. conceived the idea; N.K.T., U.M., and NBVS designed the experiments, and K.M. helped develop the NAFLD model. N.K.T. and U.M., conducted the experiments and N.K.M. and A.K.P. assisted in conducting experiments. N.K.T., U.M., T.K. and N.B.V.S. analyzed and interpreted the data; N.K.T., U.M., T.K. and N.B.V.S. wrote the paper.

## Competing interests

The authors declare no competing interests.
