## [Peer Review File · Nature Communications]

Lipid Droplet Associated Mitochondria Promotes Fatty Acid Oxidation Through Distinct Bioenergetics Pattern in Wistar male ratsREVIEWER COMMENTS

Reviewer #1 (Remarks to the Author):

In their paper, Talari et al. seek to define energetic differences in LD-associated mitochondria vs cytoplasmic mitochondria in liver. LD-associated mitochondria appear to have higher rates of fatty acid oxidation, but lower rates of succinate/malate-induced respiration. LD-associated mitochondria also have higher levels of mitofusin 2, which supports fatty acid oxidation. Overall, these studies support the existence of mitochondrial subpopulations with different energetics in the liver. However, the novelty of the approaches are overstated, and issues exist in the interpretation of the data.

- The TEM images make it extremely difficult to discern mitochondria, let alone examples of LD hydrolysis which cannot be elucidated from electron micrographs in the first place.
- The authors report “for the first time, we report the isolation of two distinct populations of mitochondria...”, but this protocol was published previously (PMID: 29617645). The authors have modified the approach slightly for rat liver, but nonetheless describe this modification as a “newly described method”. Moreover, the justification for the modification was extremely weak. The authors stated that the original method was “arduous” but provided no further explanation. The claim that this is now a novel method is grossly overstated.
- Several strong claims are made about Mfn2 knockdown affecting LD size, but this is not apparent from the images. Quantification of this effect is absent.
- The fluorescence images in Fig. 6 are totally oversaturated, making it impossible to acquire accurate reads of LD-mitochondria interactions.
- Oxygen consumption is higher in LDM compared to CM, but this is inconsistent in animal studies. For example, Fig. 7B shows an opposite effect where CM have slightly higher consumption, but this discrepancy is not addressed.

Reviewer #2 (Remarks to the Author):

In this manuscript Talari et al describe how lipid droplet associated mitochondria promote fatty acid oxidation. Based on these results, the authors suggest that this bioenergetic pattern could be used to ameliorate NAFLD. The paper is overall well organized and easy to follow.

There are several major and important points that should be addressed before this paper could be considered for publication in Nature Communications. Some of the conclusions reached by the authors are poorly supported in the current version.

1.- I find the abstract very dense and think it should be simplified containing the main findings of the paper. Also, I believe that is not common to include questions in the abstract. The revised version of the manuscript would benefit if the abstract is rewritten.

2.- I am aware that this is the first time that CM and LDM have been described, isolated and characterized as such in liver. However, these two populations of mitochondria have been deeply characterized in BAT. For this reason, the sentence included in the abstract “for the first time, we report the isolation of two distinct populations of mitochondria from rat liver” can be misleading. Maybe something like “for the first time in liver, we report the isolation of two distinct populations of mitochondria”.

3.- When describing the existence of LDM by TEM in Figure 1, the authors describe that this kind of mitochondria could have a role in hydrolysis and they illustrate that in Figure 1C. How could the authors know that those LD are undergoing hydrolysis? That conclusion should be supported when

the samples are taken after starvation and not at thermoneutrality and ad libitum to make that assumption.

4.- The authors described a new protocol to isolate LDM from liver and they analyze the different fractions obtained in the isolation to be fully characterized using imaging techniques (Figure 2). There are several concerns here that should be addressed.

4.1: In the fat layer imaging pre-stripping, there are some mitochondria in red that are not associated with LD which would mean that there is still some contamination with CM. Is there any additional fat layer washing steps to clean for potential CM contamination? How do the authors explain this observation?

4.2: Why there are no LD like structures in the post-stripped fraction? The authors showed Bodipy staining but the images look like background staining and there are no apparent LD in any of the images shown, which make the image analysis incomparable with the one from the pre-stripped fat layer.

5.- The authors then investigate FAO in LDM and CM. based in the data from Figure 1C where they claim that LDM are involved in LD hydrolysis. They find that LDM have increased fatty acid oxidation (FAO) and they associate it with increased CPT1a activity, which regulates FA import into mitochondria. Instead of addressing directly CPT1a activity, the authors look for the modulator of the CPT1a inhibitor, malonyl coA, that is regulated by ACC2. When ACC2 is phosphorylated, it inactivates ACC2 and increases FAO, and they show that p-ACC2 is higher in LDM. They should also show that CPT1a activity is increased in LDM. They should also include the reference that Malonyl Co-A is a negative regulator of CPT1a activity.

6.- When the authors describe that LDM have higher FAO but lower respiration, they should consider that the respiration presented in Figure 4 is measured using direct substrates from TCA, whereas FAO (Figure 3) is measuring using a kit. Respiration using FAO substrates should be performed as well, since by-products from FAO also enter TCA and thus ETC, contributing to respiration. It is also claimed that succinate reveals the major differences, however they show the maximal I+III activity and not II+III, supporting the results on the existence of different CoQ pools. They should also show the later, also considering that there is a great difference in Complex III activity and that the differences in Complex II is higher than in Complex I. Also, how can Complex I not be different but C I+III being so different? Are they subtracting the rotenone insensitive activity? These observations are not in accordance with the super complex model of ETC. In that respect, the authors should include better references where the different CoQ pools are described as well as super complex as respiring units.

7.- In Figure 4F, the authors are not measuring ATP synthase activity but ATP levels that are cannot be directly associated to ATP synthase activity. To measure ATP synthesis, a kinetic assay coupled to luciferin/luciferase detection in the presence of ADP and substrates and where adenylate kinase (that can also produce ATP from two molecules of ADP) should be inhibited. The labeling ATP levels in correct in the figure but not in the figure legend.

8.- Membrane potential determination in isolated mitochondria is very tricky. Is TMRE present when the imaging is being done? How is the quenching being addressed?

9.- It is unclear why the authors jump to fusion fission and mitochondrial dynamics and they focus on Mnf2. They reference again the role of Mnf2 in brown adipose tissue, where LDM were first described but I do not see how this belongs to this research line.

10.- All the experiments performed in HepG2 to test the role of Mnf2 in LDM are inconclusive because isolation of LDM from cells is very challenging and has not been proven to be effective. Also, these experiments lack the appropriate controls which make difficult to reach any conclusions. The microscopy studies performed in cells are low quality and do not allow to get specific masks that lead to appropriate segmentations to determine the LD covered by mitochondria.

11.- The quality of the blots in Supplementary Figure 1 is very poor. Complex II absent in LDM. How

can they get the quantification?

12.- Finally, the authors decide to study the role on LDM in NAFDL and for that they feed rats for 16 weeks under HFD. How can the authors explain no difference in body weight? Is there any difference in liver weight? Do they measure any other parameters of NAFDL besides serum TG? Are the liver enzymes markers affected?

13.- The claim that targeting LDM can be helpful to ameliorate NAFDL is poorly sustained. A deeper study besides bioenergetics should be provided. Are LDM levels different in SD than HFD? Are the LDM in HFD behaving like CM? How is the phosphorylation of ACC2 in CM and LDM under HFD?

Reviewer #3 (Remarks to the Author):

In this manuscript, Talari and coworkers have developed a new protocol that separates 2 types of mitochondria in rat liver: lipid-droplet associated mitochondria (LDM) and cytosolic mitochondria (CM). By using different approach they found substantial differences somehow intriguing. Under ad libitum food condition, LDM exhibited higher FAO compared to CM, albeit with lower energy production, whereas CM displayed higher bioenergetics, respiration capacity and energy production. A step further, they claim that LDM increased FAO via phospho-ACC2 and Mfn2 up-regulation. In vivo experiments in rats with HFD-induced fatty liver showed compromised FAO in liver LDM, an effect counteracted by CM that showed higher bioenergetics and respiration capacity to support expansion and increased number of LDs size and number.

Although of potential novelty and relevance in the metabolic liver disease field, this manuscript is too preliminary and many technical flaws must be amended.

Specific comments

-The first paragraph of the Results section must be moved to the introduction.

-A rationale of isolating mitochondria from rats maintained at neutral temperature (thermoneutrality?) must be provided.

-Figure 1C. Mitochondrial fission must be analyzed. Western blot analysis of mitochondrial extracts (both CM and LDM) must be analyzed for markers of mitochondrial fission (Drp, Fis1). In addition, markers of other organelles (ER, lysosomes) that also interact with LDs must be analyzed in both fractions.

-Figure 2A. The rpm must be converted into g.

-Figure 2C. The LDs seem to be broken. The authors must provide an explanation for such effect.

-In Figure 3 the authors claim that there may be an increased activity of CPT1 in LDM compared to CM. This must be demonstrated by analyzing CPT1 activity in mitochondrial extracts. Also, a control with LDM and CM mitochondria isolated from livers upon 16 h fasting must be included in this study.

-Western blots in Suppl Figure 1 and ACC2 in Figure 3B must be replaced improving the quality.

-Figure 4. Mitochondrial respiration must be evaluated using the Clark electrode. Also, the authors must consider to evaluate UCP2 in the 2 types of mitochondria.

-There are many overstatements along the manuscript. i.e. We assume that the ATP generated by CM is utilized for the anabolic processes, including FA synthesis (Figure 4F).

-Experiments in which Mfn2 is silenced in HepG2 are too preliminary and far away from the in vivo context. To fully address the role of Mfn2 in FAO by LDM, more robust methodology is needed.

Purification of LDM and CM from hepatocyte-specific Mfn2 KO mice is needed. Also, it would be very interested to conduct the studies in Drp KO mice.

-Figure S2C. An explanation for the lack of differences in body weight of HFD-fed rats must be provided.

-Figure 1. FAO must be determined in both types of mitochondria in rats fed a SD or HFD.

-The number of independent biological replicates must be specified in each panel of each Figure legend.

-The manuscript must be revised by an editing service.

REVIEWER COMMENTS

Reviewer #1 (Remarks to the Author):

In their paper, Talari et al. seek to define energetic differences in LD-associated mitochondria vs cytoplasmic mitochondria in liver. LD-associated mitochondria appear to have higher rates of fatty acid oxidation, but lower rates of succinate/malate-induced respiration. LD-associated mitochondria also have higher levels of mitofusin 2, which supports fatty acid oxidation. Overall, these studies support the existence of mitochondrial subpopulations with different energetics in the liver. However, the novelty of the approaches are overstated, and issues exist in the interpretation of the data.

Question 1

• The TEM images make it extremely difficult to discern mitochondria, let alone examples of LD hydrolysis which cannot be elucidated from electron micrographs in the first place.

Response: We agree with the Reviewer and have withdrawn the statement that mitochondria may be promoting LD hydrolysis in the revised text.

Question 2

• The authors report “for the first time, we report the isolation of two distinct populations of mitochondria...”, but this protocol was published previously (PMID: 29617645). The authors have modified the approach slightly for rat liver, but nonetheless describe this modification as a “newly described method”. Moreover, the justification for the modification was extremely weak. The authors stated that the original method was “arduous” but provided no further explanation. The claim that this is now a novel method is grossly overstated.

Response: We have taken the comment from Reviewer 1 and the suggestion from Reviewer 2 into consideration and have updated the text so that instead of ‘for the first time, we report...’ we now say, ‘for the first time in liver, we report...’

Historically, multiple attempts were made to isolate LD enriched buoyant fractions that used high speed ultracentrifugation. LDs are fragile and they fall apart during the ultracentrifugation step. We have also used the existing methods numerous times in our lab without success. In 2018, Benador et al employed low speed centrifugation and succeeded in isolating LD associated mitochondria from brown adipose tissue (BAT). Unlike BAT, that is very rich in fat, low speed centrifugation does not yield a clear fat layer from liver. Hence, one has to resort to using large number of animals that increases the work load.

Question 3

• Several strong claims are made about Mfn2 knockdown affecting LD size, but this is not apparent from the images. Quantification of this effect is absent.

Response: We thank the Reviewer for the suggestion to quantify the LD size. We have quantified the results for Figure 6A and observe that depletion of Mfn2 indeed increases the LD size by more than 3-fold.

FIG 5I in revised manuscript
Figure 5I

Question 4

•The fluorescence images in Fig. 6 are totally oversaturated, making it impossible to acquire accurate reads of LD-mitochondria interactions.

Response: *The Reviewer's observations are accurate in case of HepG2 cells treated with OA. OA treatment increases the LD size in both control and Mfn2 depletion cells, and despite several attempts, we could not further improve the resolution. Hence, we have removed Figure 6B completely and half of 6A section where OA treatment was included. We have retained the control panels of Figure 6A. As suggested by the Reviewer, we have quantified the LD sizes and have included this data as Figure 5I in the revised text.*

Question 5

•Oxygen consumption is higher in LDM compared to CM, but this is inconsistent in animal studies. For example, Fig. 7B shows an opposite effect where CM have slightly higher consumption, but this discrepancy is not addressed.

Response: *In Figure 7A (in the original manuscript), the oxygen consumption actually reflects the basal fatty acid oxidation capacity as the assay is done in the presence of carnitine, as described in the methods section. As shown, LDM out-performs CM for endogenous fatty acid oxidation activity. In case of Figure 7B, the basal oxygen consumption assay serves as a control to monitor mitochondrial respiration and does not include carnitine. We have included this information in the revised manuscript.*

Reviewer #2 (Remarks to the Author):

In this manuscript Talari et al describe how lipid droplet associated mitochondria promote fatty acid oxidation. Based on these results, the authors suggest that this bioenergetic pattern could be used to ameliorate NAFLD. The paper is overall well organized and easy to follow.

There are several major and important points that should be addressed before this paper could be considered for publication in Nature Communications. Some of the conclusions reached by the authors are poorly supported in the current version.

1.- I find the abstract very dense and think it should be simplified containing the main findings of the paper. Also, I believe that is not common to include questions in the abstract. The revised version of the manuscript would benefit if the abstract is rewritten.

Response: *We have rewritten the abstract to include only the main findings as suggested by the Reviewer in the revised manuscript and also excluded the question in it.*

2.- I am aware that this is the first time that CM and LDM have been described, isolated and characterized as such in liver. However, these two populations of mitochondria have been deeply characterized in BAT. For this reason, the sentence included in the abstract "for the first time, we report the isolation of two distinct populations of mitochondria from rat liver" can be misleading. Maybe something like "for the first time in liver, we report the isolation of two distinct populations of mitochondria".

Response: *Well noted and we have changed the text in the abstract so that it now reads, 'for the first time in liver...?'*

3.- When describing the existence of LDM by TEM in Figure 1, the authors describe that this kind of mitochondria could have a role in hydrolysis and they illustrate that in Figure 1C. How could the authors know that those LD are undergoing hydrolysis? That conclusion should be supported when the samples are taken after starvation and not at thermoneutrality and ad libitum to make that assumption.

Response: *We agree with the Reviewer and have withdrawn the statement that mitochondria may be promoting LD hydrolysis in the revised text in context of Figure 1.*

4.- The authors described a new protocol to isolate LDM from liver and they analyze the different fractions obtained in the isolation to be fully characterized using imaging techniques (Figure 2). There are several concerns here that should be addressed.

4.1: In the fat layer imaging pre-stripping, there are some mitochondria in red that are not associated with LD which would mean that there is still some contamination with CM. Is there any additional fat layer washing steps to clean for potential CM contamination? How do the authors explain this observation?

Response: *Our confocal images of LDM reveal that 95% of mitochondria are associated with LDs in the fat layer prior to pre-stripping. The chance of CM contamination is minimal as there is at least a 3ml barrier in form of buffer B between the fat layer enriched with LDM and the CM containing supernatant fraction. Any further washing decreases the LDM population significantly. We speculate that the remaining mitochondria not still in contact with LDs in the fat layer are in fact precursors to LDM. The proportion of these free mitochondria is less than 5% and we firmly believe it does not affect the outcome of the study.*

4.2: Why there are no LD like structures in the post-stripped fraction? The authors showed Bodipy staining but the images look like background staining and there are no apparent LD in any of the images shown, which make the image analysis incomparable with the one from the pre-stripped fat layer.

Response: *We have observed that while LDs are fragile, LDMs are resistant to mild vortexing. Hence, we applied mild vortexing to the pre-stripped fat layer to enrich for LDMs. Mild vortexing breaks the LDs, hence, there are no LDs visible in the post-stripped fat layer. Importantly, the pellet isolated from post-stripped fat layer has maximum LDMs.*

5.- The authors then investigate FAO in LDM and CM. based in the data from Figure 1C where they claim that LDM are involved in LD hydrolysis. They find that LDM have increased fatty acid oxidation (FAO) and they associate it with increased CPT1a activity, which regulates FA import into mitochondria. Instead of addressing directly CPT1a activity, the authors look for the modulator of the CPT1a inhibitor, malonyl coA, that is regulated by ACC2. When ACC2 is phosphorylated, it inactivates ACC2 and increases FAO, and they show that p-ACC2 is higher in LDM. They should also show that CPT1a activity is increased in LDM. They should also include the reference that Malonyl Co-A is a negative regulator of CPT1a activity.

Response: *A reference has been included that implicates Malonyl CoA as a negative regulator of CPT1a activity in the revised manuscript (McGarry JD, Brown NF 1997). We have also carried out the CPT1 activity assay and the results are included in the revised manuscript (Figure 3D) and also shown below. Consistent with our previous findings and as hypothesized, LDM has relatively higher CPT1 activity compared to CM.*

FIG 3D in the revised manuscript

Figure 3D

6.- When the authors describe that LDM have higher FAO but lower respiration, they should consider that the respiration presented in Figure 4 is measured using direct substrates from TCA, whereas FAO (Figure 3) is measuring using a kit. Respiration using FAO substrates should be performed as well, since by-products from FAO also enter TCA and thus ETC, contributing to respiration. It is also claimed that

succinate reveals the major differences, however they show the maximal I+III activity and not II+III, supporting the results on the existence of different CoQ pools. They should also show the later, also considering that there is a great difference in Complex III activity and that the differences in Complex II is higher than in Complex I. Also, how can Complex I not be different but C I+III being so different? Are they subtracting the rotenone insensitive activity? These observations are not in accordance with the super complex model of ETC. In that respect, the authors should include better references where the different CoQ pools are described as well as super complex as respiring units.

Response: In FAO assay, along with the FAO substrate oleate, carnitine is included to stimulate FAO. In case of mitochondrial respiration assay, assay is performed with TCA substrates like succinate or glutamate/malate. However, the final read out in both the assays is same which is oxygen consumption. Hence, FAO can be considered as mitochondrial respiration in the presence of FAO substrate.

We have monitored the complex II+III activity in presence of succinate and the results are included in the revised text and also shown below. Complex II+III activity is compromised in LDM compared to CM. These results are consistent with our previous observations when we monitored Complex I+III activity.

FIG 4F in revised manuscript

We do not see statistically significant differences in the individual complex activities between LDM and CM.

The Reviewer is accurate in noting that though individual complex activities are comparable, CI+CIII activity is different. We attribute this difference to the following reasons: a) When carrying out individual complex activities, unlimited amount of CoQ is added. However, in case of CI+CIII assay, no CoQ is added. The assay is dependent on the endogenous pool of CoQ. b) We believe that the endogenous pool of CoQ might be low in LDM compared to CM. c) It is possible that LDM and CM differ in the super-complex respiring pools as the Reviewer has suggested.

We haven't subtracted rotenone insensitive activity.

7.- In Figure 4F, the authors are not measuring ATP synthase activity but ATP levels that are cannot be directly associated to ATP synthase activity. To measure ATP synthesis, a kinetic assay coupled to luciferin/luciferase detection in the presence of ADP and substrates and where adenylate kinase (that can also produce ATP from two molecules of ADP) should be inhibited. The labeling ATP levels in correct in the figure but not in the figure legend.

Response: The labeling in the figure legend has been corrected..

8.- Membrane potential determination in isolated mitochondria is very tricky. Is TMRE present when the imaging is being done? How is the quenching being addressed?

Response: Yes, TMRE is present while imaging. We did not have any issue with quenching. Both mitochondrial fractions were incubated with equal amount of TMRE for exact length of time.

9.- It is unclear why the authors jump to fusion fission and mitochondrial dynamics and they focus on Mfn2. They reference again the role of Mfn2 in brown adipose tissue, where LDM were first described but I do not see how this belongs to this research line.

Response: *Our laboratory is very interested in mitochondrial dynamics. Please see our recent paper (JBC,2022; <https://pubmed.ncbi.nlm.nih.gov/36162502>). In case of liver, there are no previous studies on LDM. We had to rely on studies that were conducted in brown adipose tissue. One study demonstrated that Mfn2 mediates the interaction between LD and mitochondria to regulate lipolysis (EMBO Journal, 2017) (<https://pubmed.ncbi.nlm.nih.gov/28348166/>). In another study, Mfn2 depletion impairs thermogenesis and increases LD size (EMRO Reports,2017) (<https://pubmed.ncbi.nlm.nih.gov/28539390/>). More recently, Benador et al showed increased Mfn2 expression in peridroplet mitochondria (PDM) compared to CM. In addition, they observed that PDM supports LD expansion, while CM boosts FAO. Taking into consideration the above findings, we examined Mfn2.*

10.- All the experiments performed in HepG2 to test the role of Mfn2 in LDM are inconclusive because isolation of LDM from cells is very challenging and has not been proven to be effective. Also, these experiments lack the appropriate controls which make difficult to reach any conclusions. The microscopy studies performed in cells are low quality and do not allow to get specific masks that lead to appropriate segmentations to determine the LD covered by mitochondria.

Response: *The Reviewer's observations are accurate in case of HepG2 cells treated with OA. OA treatment increases the LD size in both control and Mfn2 depletion cells, and despite several attempts, we could not further improve the resolution. Hence, we have removed Figure 6B completely and half of 6A section where OA treatment was included. We have retained the control panels of Figure 6A. As suggested by Reviewer I, we have quantified the LD sizes and have included this data as Figure 5I in the revised text.*

FIG 5I in the revised manuscript

Figure 5I

11.- The quality of the blots in Supplementary Figure 1 is very poor. Complex II absent in LDM. How can they get the quantification?

Response: *Well noted. We have repeated the experiment to obtain a much-improved blot. This replaces the original blot. Please see below:*

FIG. S1A and B in the revised manuscript

12.- Finally, the authors decide to study the role on LDM in NAFDL and for that they feed rats for 16 weeks under HFD. How can the authors explain no difference in body weight? Is there any difference in liver weight? Do they measure any other parameters of NAFDL besides serum TG? Are the liver enzymes markers affected?

Response: Though we did not see a difference in body weight except at the 8th week (Figure S2C), the HFD fed animals displayed all the characteristics of NAFLD. These included increased serum triglycerides (Figure S2H), hepatic steatosis in H&E liver sections (Figure S2F), and morphological changes (Figure S2A). We have now examined levels of serum total cholesterol and it has been included in the revised manuscript (Figure S2G). In HFD fed animals, the serum total cholesterol is increased.

FIG S2G in the revised manuscript

Figure S2G

As suggested by the Reviewer, we have reviewed the weights of the liver documented at the time of animal sacrifice. Indeed, the livers from the HFD fed animals had gained significantly more weight (Figure S2 D in the revised manuscript), and liver/BW ratio shows significant increase in HFD fed animals (Figure S2 E) however, curiously, this is not apparent in the total body weight. We have no explanation for this at this point in time.

FIG S2D and E in the revised manuscript

13.- The claim that targeting LDM can be helpful to ameliorate NAFLD is poorly sustained. A deeper study besides bioenergetics should be provided. Are LDM levels different in SD than HFD? Are the LDM in HFD behaving like CM? How is the phosphorylation of ACC2 in CM and LDM under HFD?

Response: In liver, this is the first study to report the existence of two distinct sub-populations of mitochondria. Within the scope of this study, we have delineated the differences between LDM and CM both bioenergetically and biochemically. We have observed that the levels of LDM increases in NAFLD condition. The total LDM pellet is on an average of 150-200ug/animal in SD, whereas it is 250-300ug/animal in HFD group. This indicates that the liver is trying to compensate for the reduced FAO function of LDM in NAFLD condition by increasing the mass of LDM. Nevertheless, this is not enough to rescue the liver from the HFD as evident from the extent of steatosis that has already set in. In HFD, the LDM is not able to carry out its specialized function and its bioenergetics pattern appear to close in to that of CM. This conclusion is remarkably reinforced when we look at pACC2/ACC2 levels as suggested by the Reviewer. The hallmark of LDM, pACC2, is dramatically reduced in NAFLD condition (Figure 6E in the revised manuscript) and ACC2 levels are also decreased. Both bioenergetically and biochemically, we show that LDM has a specialized function and this function is compromised in NAFLD condition. We believe our study provides novel insights on liver mitochondria, the distinct mitochondrial sub-populations and how LDM can potentially serve as a target for therapeutic interventions for the amelioration of NAFLD.

Figure 6E in the revised manuscript

Figure 6E

Reviewer #3 (Remarks to the Author):

In this manuscript, Talari and coworkers have developed a new protocol that separates 2 types of mitochondria in rat liver: lipid-droplet associated mitochondria (LDM) and cytosolic mitochondria (CM). By using different approach they found substantial differences somehow intriguing. Under ad libitum food condition, LDM exhibited higher FAO compared to CM, albeit with lower energy production, whereas CM displayed higher bioenergetics, respiration capacity and energy production. A step further, they claim that LDM increased FAO via phospho-ACC2 and Mfn2 up-regulation. In vivo experiments in rats with HFD-induced fatty liver showed compromised FAO in liver LDM, an effect counteracted by CM that showed higher bioenergetics and respiration capacity to support expansion and increased number of LDs size and number.

Although of potential novelty and relevance in the metabolic liver disease field, this manuscript is too preliminary and many technical flaws must be amended.

Specific comments

-The first paragraph of the Results section must be moved to the introduction.

Response: *Noted and part of first paragraph of the results section moved to the introduction..*

-A rationale of isolating mitochondria from rats maintained at neutral temperature (thermoneutrality?) must be provided.

Response: *It is a misnomer in the legend to Figure 1, the animals were kept at room temperature as mentioned in the methods. The legend has been corrected.*

-Figure 1C. Mitochondrial fission must be analyzed. Western blot analysis of mitochondrial extracts (both CM and LDM) must be analyzed for markers of mitochondrial fission (Drp, Fis1). In addition, markers of other organelles (ER, lysosomes) that also interact with LDs must be analyzed in both fractions.

Response: As suggested by the Reviewer, we have carried out western blot analysis of LDM and CM using Drp1 and phosphor-Drp1 antibodies. Despite comparable amount of Drp1 in both sets of mitochondria, we find that CM has relatively higher phosphor-Drp1 presence compared to LDM as shown below.

FIG S1 D and E in the revised manuscript

In addition, we have also probed the LDM and CM samples with other antibodies for detecting presence of ER (GRP78 and calreticulin), lysosomes (lamp1) and cytoplasm (GAPDH) as shown below. Both CM and LDM show presence of ER while mild lysosomal presence is observed in LDM sample. However, both CM and LDM are negative for cytoplasm. Further in-depth studies are required to unravel the importance of inter-organelle communication on the mitochondrial sub-populations which is beyond the scope of this study.

FIG S2 C in the revised manuscript

Figure S1C

-Figure 2A. The rpm must be converted into g.

Response: Noted and converted.

-Figure 2C. The LDs seem to be broken. The authors must provide an explanation for such effect.

Response: We have observed that while LDs are fragile, LDMs are resistant to mild vortexing. Hence, we applied mild vortexing to the pre-stripped fat layer to enrich for LDMs. Mild vortexing breaks the LDs, hence, there are no LDs visible in the post-stripped fat layer. Importantly, the pellet from the post-stripped fat layer has maximum LDMs.

-In Figure 3 the authors claim that there may be an increased activity of CPT1 in LDM compared to CM. This must be demonstrated by analyzing CPT1 activity in mitochondrial extracts. Also, a control with LDM and CM mitochondria isolated from livers upon 16 h fasting must be included in this study.

Response: We have carried out the CPT1 activity assay and the results are included in the revised manuscript (Figure 3D) and also shown below. Consistent with our previous findings and as hypothesized, LDM has relatively higher CPT1 activity compared to CM.

Figure 3D in the revised manuscript

Figure 3D

We are not able to isolate liver LDM from one fasting animal, hence, we could not include this sample while performing the CPT1 assay. However, we checked FAO activity in LDM and CM samples isolated from pooled liver harvested from 4 fasting animals as shown below. For reference, we used LDM and CM sample isolated from 3 animals fed with ad libitum diet. Consistent with our earlier findings, LDM has higher FAO activity than CM in control ad libitum condition. However, the FAO activity of both liver LDM and CM from fasting animals are comparable and importantly, their values are similar to control LDM. During fasting, CM is showing higher FAO to compensate for the lack of food. As we did not observe any change in FAO in LDM samples isolated from fasting animals and ad libitum-fed animals, we did not perform CPT1 activity.

-Western blots in Suppl Figure 1 and ACC2 in Figure 3B must be replaced improving the quality.

Response: We have repeated the supplementary Figure 1 and Figure 3B. We have replaced the supplementary Figure 1 with a better-quality figure in the revised manuscript. In case of Figure 3B, we retained the original figure as we could not further improve its quality.

-Figure 4. Mitochondrial respiration must be evaluated using the Clark electrode. Also, the authors must consider to evaluate UCP2 in the 2 types of mitochondria.

Response: We are unable to evaluate mitochondrial respiration using the Clark electrode as the assay requires large amount of mitochondrial sample and the LDM fraction is very low.

We examined the expression of UCP2 in LDM and CM samples and find that its expression is reduced or nearly abolished in LDM sample as shown below. This data has been included in the revised manuscript (FigureS1A).

FIG S1A and B in the revised manuscript

-There are many overstatements along the manuscript. i.e. We assume that the ATP generated by CM is utilized for the anabolic processes, including FA synthesis (Figure 4F).

Response: As said, it is purely an assumption. Nevertheless, we have now removed this statement.

-Experiments in which Mfn2 is silenced in HepG2 are too preliminary and far away from the in vivo context. To fully address the role of Mfn2 in FAO by LDM, more robust methodology is needed. Purification of LDM and CM from hepatocyte-specific Mfn2 KO mice is needed. Also, it would be very interesting to conduct the studies in Drp KO mice.

Response: As explained in the manuscript, we are unable to isolate LDM from cell lines. It is beyond the scope of this study to include studies on KO mice.

-Figure S2C. An explanation for the lack of differences in body weight of HFD-fed rats must be provided.

Response: Though we did not see a difference in body weight except at the 8th week (Figure S2C), the HFD fed animals displayed all the characteristics of NAFLD. These included increased serum triglycerides (Figure S2H), hepatic steatosis in H&E liver sections (Figure S2F), and morphological changes (Figure S2A). We have now examined levels of serum total cholesterol and it has been included in the revised manuscript (Figure S2G). In HFD fed animals, the serum total cholesterol is increased.

FIG S2G in the revised manuscript

Figure S2G

We have reviewed the weights of the liver documented at the time of animal sacrifice. The livers from the HFD fed animals had gained significantly more weight (Figure S2 D in the revised manuscript), and liver/BW ratio shows significant increase in HFD fed animals (Figure S2 E) however, curiously, this is not apparent in the total body weight. We have no explanation for this at this point in time.

FIG S2 D and E in the revised manuscript

Figure S2D

Figure S2E

-Figure 1. FAO must be determined in both types of mitochondria in rats fed a SD or HFD.

Response: We would like to draw the Reviewer's attention to Figure 7A. The read out in FAO assay is oxygen consumption. Hence, the oxygen consumption shown in Figure 7A is FAO as it is done in presence of oleate and carnitine. To avoid confusion with the respiration assay where the read out is oxygen consumption in presence of TCA substrates and ADP, we are now replacing the 'oxygen consumption' on the Y axis in FAO assay with 'FAO capacity'.

-The number of independent biological replicates must be specified in each panel of each Figure legend.

Response: We have specified the number of independent biological replicates in the revised manuscript.

REVIEWER COMMENTS

Reviewer #1 (Remarks to the Author):

The authors have adequately addressed my initial concerns. Although the Mfn2 studies are a little disjointed in my opinion, the major findings that liver mitochondria exist within two subpopulations (LDM and CM) and exhibit distinct energetics is convincing. The manuscript is improved in its description of key methodologies and description of the the relative novelty with respect to previous studies.

Reviewer #2 (Remarks to the Author):

In this revised version of the manuscript Talari et al have addressed most of my comments from the first round of revisions. However, there are still some minor points that should be addressed before the manuscript is accepted for publication in Nature Communications.

Regarding the methodology to isolate LDM, the authors have clarified all the points addressed in my previous review. However, I still have a question and a suggestion. The question is: why do they vortex the samples to detach the LDM from the LD? Have they tried without vortexing? LDM should detach by differential centrifugation. Since it is the first time that this protocol is described for liver, they should comment on that since for BAT no vortexing is needed to detach mitochondria from LD. In addition, since this manuscript is the first one describing the separation of two distinctive mitochondrial populations in liver, the reader would benefit not only from the scheme shown in Figure 2A, but also from real pictures from an isolation to see how the fractions look when performing the experiment. This should be either added as Figure 2A instead of the scheme or as an additional panel in Supplementary.

In line 94, the authors talk about nutrient starvation. Liver responds very differently metabolically depending if there is fasting or starvation, activating either gluconeogenesis or lipolysis. This point should be clarified in the text.

In Figure 1, the authors have beautifully indicated what correspond to LD and what to mitochondria in green and red, respectively. They have also added the description in the figure legend. I would suggest to also add what green and red refer to in the Figure itself.

When referring to measuring the activities of the individual and combined ETC complexes, the authors make a reasonable point. However, when addressing Complex I activity, it should always be taken into account how much of the activity detected is sensitive to rotenone. Also, the authors should include references referring to SC and CoQ pools. Here are some suggestions: PMID: 19026783, PMID: 23812712, PMID: 25126045

I also have noted that the references are not all of them in the same format. Some of the references start with the last name and their first name initial (example: Smith, L et al...) while other start with the first name and then their last name (Samuel Smith et al...). Formatting should be consistent.

Reviewer #3 (Remarks to the Author):

This revised version has improved in some aspects regarding the protocol for isolating LDM and CM and the characterization of the differences in bioenergetics/functional properties. However, besides the novelty in methodology, this study needs more in depth research. Regarding the previous comments, I still consider that the use of HepG2 cells to unravel the role of mitofusin in LDM is not conclusive. Due to the potential novelty of this issue, these set of experiments must be conducted in non-transformed hepatocytes and in vivo. Also, the manuscript still needs English editing to avoid statements such as “the newly described method, one can cleanly separate and isolate LDM from adult rat liver” (line 127).

REVIEWER COMMENTS

Reviewer #1 (Remarks to the Author):

The authors have adequately addressed my initial concerns. Although the Mfn2 studies are a little disjointed in my opinion, the major findings that liver mitochondria exist within two subpopulations (LDM and CM) and exhibit distinct energetics is convincing. The manuscript is improved in its description of key methodologies and description of the the relative novelty with respect to previous studies.

Response: *We thank the Reviewer for his comments. Taking into consideration of your comments on the Mfn2 data (Figure 5), we have moved the Mfn2 cell line data to the supplementary section (Supplementary Figure S3 in the revised manuscript).*

Reviewer #2 (Remarks to the Author):

In this revised version of the manuscript Talari et al have addressed most of my comments from the first round of revisions. However, there are still some minor points that should be addressed before the manuscript is accepted for publication in Nature Communications.

Regarding the methodology to isolate LDM, the authors have clarified all the points addressed in my previous review. However, I still have a question and a suggestion. The question is: why do they vortex the samples to detach the LDM from the LD? Have they tried without vortexing? LDM should detach by differential centrifugation. Since it is the first time that this protocol is described for liver, they should comment on that since for BAT no vortexing is needed to detach mitochondria from LD. In addition, since this manuscript is the first one describing the separation of two distinctive mitochondrial populations in liver, the reader would benefit not only from the scheme shown in Figure 2A, but also from real pictures from an isolation to see how the fractions look when performing the experiment. This should be either added as Figure 2A instead of the scheme or as an additional panel in Supplementary.

Response: *We have indeed tried to isolate LDM without vortexing, but we failed to dissociate the LDs from mitochondria. Vortexing helped to loosen the LD-mitochondrial contacts, which we now mentioned in the Results section. We have also included the pictures taken during the LDM isolation in the Supplementary section (Supplementary Figure1).*

In line 94, the authors talk about nutrient starvation. Liver responds very differently metabolically depending if there is fasting or starvation, activating either gluconeogenesis or lipolysis. This point should be clarified in the text.

Response: *Noted and the text has been modified and highlighted in the revised manuscript.*

In Figure 1, the authors have beautifully indicated what correspond to LD and what to mitochondria in green and red, respectively. They have also added the description in the figure legend. I would suggest to also add what green and red refer to in the Figure itself.

Response: *We thank the Reviewer for the kind comment. We have accordingly inserted the legend in the Figure too.*

When referring to measuring the activities of the individual and combined ETC complexes, the authors make a reasonable point. However, when addressing Complex I activity, it should always be taken into account how much of the activity detected is sensitive to rotenone. Also, the authors should include references referring to SC and CoQ pools. Here are some suggestions: PMID: 19026783, PMID: 23812712, PMID: 25126045

Response: *Initially, we carried out Complex I activity in the presence of rotenone and observed that more than 90% of the observed activity is rotenone sensitive. Hence, we carried out further experiments in the absence of rotenone. We thank the Reviewer for the References, and we have included these in the revised manuscript.*

I also have noted that the references are not all of them in the same format. Some of the references start with the last name and their first name initial (example: Smith, L et al...) while other start with the first name and then their last name (Samuel Smith et al...). Formatting should be consistent.

Response: *We thank the Reviewer for the observation and have re-formatted all the References according to the guidelines.*

Reviewer #3 (Remarks to the Author):

This revised version has improved in some aspects regarding the protocol for isolating LDM and CM and the characterization of the differences in bioenergetics/functional properties. However, besides the novelty in methodology, this study needs more in depth research. Regarding the previous comments, I still consider that the use of HepG2 cells to unravel the role of mitofusin in LDM is not conclusive. Due to the potential novelty of this issue, these set of experiments must be conducted in non-transformed hepatocytes and in vivo. Also, the manuscript still needs English editing to avoid statements such as “the newly described method, one can cleanly separate and isolate LDM from adult rat liver” (line 127).

Response: *We thank the Reviewer for his comments. Our results show that the overall depletion of Mfn2 affects lipid oxidation and LDs size in HepG2 cells. However, the precise role of Mfn2 can only be delineated in LDMs once an efficient method is discovered to isolate LDMs from transformed or non-transformed cell lines or primary hepatocytes, which are technically challenging, as reviewer 2 pointed out. As pointed out by the reviewer 1, reviewer 2 in the initial version and based on reviewer 3 comments, we felt that Mfn2 data might be better suitable in the Supplementary section. Hence, we have moved the Mfn2 cell line data to the supplementary section (Supplementary Figure 3) and modified the manuscript accordingly.*

We have carefully reviewed the manuscript and edited it as the Reviewer suggested.